# Sodium nitrite promotes atherosclerosis via IL-1β: Network toxicology and machine learning insights

**HaoBo Yang**[1,2], **YunFeng Yu**[1,2], **YongHui Zhang**[1,2], **YaNan Bai**[1,2], **YaRu Shi**[1,2], **WeiXiong Jian**[1,2]*

1 Hunan Provincial Key Laboratory of TCM Diagnostics, Institute of National Key Discipline in TCM Diagnostics, Changsha, Hunan, China, 2 Hunan University of Chinese Medicine, Changsha, Hunan, China

* daxiong20001977@163.com

## Abstract

### Background

Atherosclerosis (AS) is a major global health burden. Sodium nitrite, a common environmental and dietary contaminant, has been implicated in promoting AS, but its systematic molecular mechanisms remain unclear.

### Methods and results

To address this gap, we integrated network toxicology, machine learning, transcriptomics, molecular docking, and molecular dynamics simulations. Disease-related targets were first identified from public databases, and four core candidates—IL-1β, IL6, PTK2, and NOS3—were prioritized using machine learning approaches. Molecular docking confirmed strong and stable binding affinities between sodium nitrite and these targets, while molecular dynamics simulations further validated the stability of the sodium nitrite–IL-1β complex. Moreover, immune infiltration analysis revealed a significant increase in monocyte/macrophage infiltration within AS plaque tissues, suggesting that sodium nitrite–related targets are associated with immune microenvironment changes in AS.

### Conclusion

Collectively, this study proposed an adverse outcome pathway (AOP) framework linking sodium nitrite exposure to atherosclerosis, providing hypothesis-generating mechanistic insights for future experimental validation.

**Data availability statement:** Public datasets analyzed in this study are available at GEO: GSE28829 (https://www.ncbi.nlm.nih.gov/geo/query/acc.cgi?acc=GSE28829) and GSE100927 (https://www.ncbi.nlm.nih.gov/geo/query/acc.cgi?acc=GSE100927).

**Funding:** This work was supported by the following: National Natural Science Foundation of China (Grant No. 82374334). URL: https://www.nsfc.gov.cn/ Natural Science Foundation of Hunan Province (Grant No. 2024JJ9466). URL: http://kjt.hunan.gov.cn/ Graduate Innovation Project of Hunan Province (Grant No. CX20251169). URL: http://jyt.hunan.gov.cn/ The funders had no role in study design, data collection and analysis, decision to publish, or preparation of the manuscript.

**Competing interests:** The authors have declared that no competing interests exist.

## Introduction

Atherosclerosis (AS) and its complications, including myocardial infarction and stroke, represent a major global health burden. The disease is shaped by multiple factors, such as elevated blood lipids, vascular endothelial dysfunction, and inflammatory cell infiltration. Its progression is primarily driven by four interrelated elements: immune system activity, inflammatory processes, lipid metabolism abnormalities, and traditional cardiovascular risk factors. The interactions among these elements highlight the multifactorial nature of atherosclerosis, with each factor contributing through distinct mechanisms and pathways that collectively accelerate disease development [1].

The relationship between sodium nitrite and AS is bidirectional. Clinically, sodium nitrite has mainly been used as an antidote under acute [2], closely monitored conditions, and its hemodynamic effects have been evaluated in controlled short-term infusion studies [3].In the human body, the toxicity of sodium nitrite is often associated with high-dose exposure, especially in suicidal sodium nitrite ingestion, where the nitrite concentration in the blood may reach 4.32 mM and high concentrations can cause the formation of high levels of MetHb, leading to hypoxia and multi organ damage [4].In experimental models and early-phase human studies, nitrite (including inhaled sodium nitrite formulations) has been evaluated as a nitric oxide donor capable of acutely lowering pulmonary pressures in pulmonary hypertension and as a cytoprotective agent in ischemia–reperfusion settings [5].At low doses, sodium nitrite may exert protective effects through nitric oxide (NO)-related pathways, whereas high doses or inappropriate exposure can accelerate disease progression by inducing oxidative stress and endothelial injury. Upon metabolism, sodium nitrite is converted into NO, which subsequently mitigates the effects of reactive oxygen species, such as superoxide anions ($O_2^-$). This action helps reduce the oxidative modification of low-density lipoproteins (LDL), which is a critical step in preventing the formation of oxidized LDL (ox-LDL). Because ox-LDL is a critical driver of lipid core development within atherosclerotic plaques, the protective effect of sodium nitrite at low levels is closely linked to its ability to limit LDL oxidation [6]. At low doses or during short-term exposure, sodium nitrite is generally considered protective and exhibits minimal toxicity [7,8]. Studies have demonstrated that sodium nitrite can be converted into NOin the gastric acidic environment. NO subsequently activates the sGC–cGMP pathway, leading to relaxation of vascular smooth muscle, improved blood flow, and reduced blood pressure. In addition, NO inhibits platelet glycoprotein IIb/IIIa receptors, thereby lowering the risk of thrombosis.

In contrast, high-dose or long-term exposure to sodium nitrite has detrimental effects. Experimental studies have shown that chronic exposure accelerates collagen deposition in the arterial wall, increases vascular stiffness, and promotes the generation of reactive nitrogen species (RNS) such as peroxynitrite. These RNS attack endothelial cells, trigger lipid peroxidation, and stimulate the release of pro-inflammatory cytokines including IL-6 and TNF-α. Moreover, sodium nitrite can induce methemoglobinemia, which reduces oxygen delivery and causes hypoxic injury to the vascular endothelium [9].Sodium nitrite can also react with amines to form nitrosamines, which

are classified as group 2A carcinogens. These compounds contribute to chronic inflammation and DNA damage, thereby indirectly promoting atherosclerotic plaque formation [10].

Sodium nitrite is widely present in daily life, with food consumption representing the most common exposure route. It is commonly added to processed meats such as ham, sausages, cured meats, and bacon, where it helps retain red color, inhibit microbial growth (particularly Clostridium botulinum), and enhance flavor. Beyond the food industry, sodium nitrite is also used in textile dyeing, metal corrosion inhibition, and bleaching processes. In economically disadvantaged regions, contaminated groundwater and polluted water sources may contain elevated levels of nitrate/nitrite, and long-term consumption of such water markedly increases exposure risk.

## Methods

### Collection of sodium nitrite targets

The chemical structure and simplified molecular input line entry system (SMILES) representation of sodium nitrite were retrieved from the PubChem database (https://pubchem.ncbi.nlm.nih.gov/). Potential sodium nitrite-related targets were then collected from ChEMBL (https://www.ebi.ac.uk/chembl/), STITCH (http://stitch.embl.de/), and the Comparative Toxicogenomics Database (CTD, https://ctdbase.org/), with the search limited to Homo sapiens. After compilation, duplicate entries were excluded, and the remaining unique targets were retained for further analysis.

### Acquisition of potential atherosclerosis-related targets

AS-related targets were sourced from the Online Mendelian Inheritance in Man (OMIM, https://omim.org/) and GeneCards (https://www.genecards.org/) databases. The retrieved targets were integrated, duplicate entries were removed, and a final list of potential AS-associated genes was established for further analysis. To balance comprehensiveness with analytical manageability, genes retrieved from GeneCards were ranked by their relevance scores. A threshold of relevance score ≥ 1.26 was applied to ensure the inclusion of genes with substantial association evidence. This threshold yielded the top 2000 ranked genes, which were selected for subsequent analysis. This approach was designed to capture a robust set of high-confidence candidate genes while maintaining a tractable dataset size for downstream network and machine learning analyses. In the case of STITCH, only interactions with a confidence score ≥ 0.7 were considered.

### Functional pathway analysis of target genes

Functional enrichment analysis was conducted using the DAVID database (https://david.ncifcrf.gov/). Gene Ontology (GO) and Kyoto Encyclopedia of Genes and Genomes (KEGG) analyses were performed to identify molecular pathways associated with sodium nitrite-induced AS targets. The analysis included the three primary GO categories—Biological Process, Cellular Component, and Molecular Function—along with KEGG pathways. Pathways or functions with adjusted p-values < 0.05 were considered to be significantly enriched.

### Screening of core target genes using machine learning algorithms

The purpose of this machine learning (ML) step was to perform feature prioritization and refinement within a focused set of candidate targets. Specifically, the 24 overlapping sodium nitrite-AS candidate targets constituting the feature matrix were derived from the preceding network toxicology and database mining analyses (Sections 2.1–2.3), representing genes with prior biological plausibility for involvement in the mechanism of interest. Therefore, the application of LASSO and SVM-RFE was not intended for high-dimensional biomarker discovery from a full transcriptome, but rather to identify the most critical core drivers from this pre-filtered, biologically relevant candidate list. This approach of applying feature selection to a biologically curated gene set is a common strategy to enhance model interpretability and mitigate the risk of overfitting that can occur with high-dimensional data. The feature matrix for machine learning consisted of the expression values of

24 candidate genes in the GSE28829 dataset (n = 29; AS=16, control = 13). These 24 genes represent the overlapping targets between sodium nitrite-related genes and AS-associated genes, as identified through network toxicology analysis. Thus, the input features were pre-filtered based on biological plausibility, with a total of 24 features included in the initial model. Two machine learning algorithms were employed to perform feature selection and identify core genes associated with sodium nitrite-induced AS: LASSO regression and SVM-RFE. LASSO regression integrates built-in feature selection with L1 regularization, shrinking the coefficients of less important features to zero and thereby selecting a subset of predictive genes. SVM-RFE is a wrapper-based feature selection method that iteratively removes the least important features based on SVM weight coefficients, identifying the optimal feature subset through cross-validation. Both methods were applied to the 24-feature matrix to further refine the gene set.Core target genes were defined as the intersection of genes identified by both algorithms, resulting in a final set of four features (IL-1β, IL6, PTK2, and NOS3) that were considered the most robust candidates for subsequent analyses. The implementation details were as follows: LASSO was performed using the glmnet R package with the parameters standardize = TRUE, alpha = 1, family = "binomial", nfolds = 10. For SVM-RFE, feature ranking was implemented using the e1071 package (svm function) with a linear kernel (kernel = "linear", type = "C-classification"). The main parameters were set as cost (C) = 10 and cachesize = 500.Prior to SVM-RFE, the expression matrix was standardized; therefore, internal scaling in SVM fitting was disabled (scale = FALSE). Regular SVM-RFE was applied (k = 1), and when the number of remaining features exceeded 5000, features were removed by halving at each iteration (halve.above = 5000); otherwise, features were eliminated sequentially. For LASSO regression, the analysis was performed using the glmnetR package. The key parameters were set as follows: family = "binomial"for binary classification, alpha = 1to apply the L1 (LASSO) penalty, and standardize = TRUEto standardize features before model fitting. The optimal regularization parameter (λ) was determined via 10-fold cross-validation (nfolds = 10), with the specific λ value that minimized the cross-validation deviance (lambda.min) selected for the final model. A fixed random seed (e.g., set.seed = 123) was employed in the cross-validation procedures to ensure reproducibility.

For internal validation, a 10-fold cross-validation strategy was employed. The GSE28829 dataset was randomly partitioned into 10 approximately equal-sized subsets. In each iteration, nine subsets were used for model training and the remaining subset for validation, with the process repeated ten times to ensure all samples were used for validation once. The average performance metrics across folds were reported. For external validation, the fully trained model (using all GSE28829 samples) was applied to an independent dataset, GSE100927. This independent evaluation assesses the generalizability of the identified core genes.

To assess the robustness of the Lasso regression model, we included performance metrics such as AUC, precision, recall, and F1-score. In addition to the 10-fold cross-validation for model tuning and lambda selection, an independent external validation was performed using the GSE100927 dataset. The diagnostic performance was evaluated using ROC curves, and the AUC values were calculated for each of the core genes. Furthermore, classification performance was evaluated using precision, recall, F1-score, and the Youden index to assess sensitivity and specificity.

### Differential expression and diagnostic evaluation of core genes and validation with external datasets

The GEO dataset GSE28829, which contains gene expression data from AS plaque tissues and normal samples, was used to evaluate the expression of the core genes. Genes with p-values < 0.05 were considered significantly differentially expressed. Based on these expression profiles, receiver operating characteristic (ROC) curves were generated to assess the diagnostic potential of the identified core targets, and diagnostic accuracy was quantified by the area under the curve (AUC).

External validation was conducted using GSE100927. ROC curves were generated with the pROC package, and AUCs with 95% confidence intervals were calculated using DeLong's method. The optimal cutoff was determined by the Youden index, and sensitivity and specificity were reported.

## Immune infiltration analysis

Single-sample Gene Set Enrichment Analysis (ssGSEA) was conducted using the GSVA R package (v1.46.0) to evaluate immune cell infiltration in AS plaque tissues compared with normal tissues. Statistical significance was assessed using the Wilcoxon rank-sum test [11]. Correlations between core target gene expression and immune cell infiltration were also evaluated [12].

## Molecular docking

The three-dimensional structures of the core target proteins were retrieved from the Protein Data Bank (PDB, http://www.rcsb.org/), and the molecular structure of sodium nitrite was obtained from PubChem (http://pubchem.ncbi.nlm.nih.gov/). Structure files were converted into Mol2 format using Open Babel. Protein preprocessing involved the removal of water molecules and the addition of hydrogen atoms. Molecular docking was performed using AutoDockTools 1.5.7, and docking results generated by AutoDock Vina (in pdbqt format) were converted to pdb format using PyMOL. PyMOL 3.2 was then utilized to visualize the protein–ligand complexes with the highest binding affinities [13]. Following 3D visualization, 2D diagrams depicting the specific molecular interactions between sodium nitrite and the target proteins were generated using LigPlot+. These diagrams provide a schematic representation of the binding interactions for further analysis.

## Molecular dynamics simulation

Molecular dynamics (MD) simulations of the sodium nitrite–interleukin-1β (IL-1β) complex were conducted using Gromacs 2022.3. The small molecule was preprocessed with AmberTools 22: atom types were assigned using the General Amber Force Field (GAFF), hydrogen atoms were added, and restrained electrostatic potential (RESP) charges were calculated using Gaussian 16W. The calculated charges were then incorporated into the topology file for the simulation [14]. Simulations were performed under constant temperature (300 K) and pressure (1 bar) for a total duration of 100 ns. The CHARMM36 force field was used to model biomolecular interactions, the system was solvated with the TIP3P water model, and $Na^+$ ions were added to neutralize the overall charge. The simulation protocol included steepest descent energy minimization to optimize the initial structure, equilibration to stabilize system parameters, and a 100 ns production run to sample conformational dynamics.

## Target phenotypes and gene–phenotype network

Phenotypes strongly associated with sodium nitrite and AS were identified as potential key events (KEs), and the genes linked to these phenotypes were defined as molecular initiating events (MIEs). By integrating hierarchical relationships and biological associations reported in the literature, an Adverse Outcome Pathway (AOP) was constructed to link sodium nitrite exposure with the pathogenesis of AS.

# Results

## Network toxicology analysis of potential sodium nitrite targets in atherosclerosis

Network toxicology, an advanced approach for predicting the mechanisms and targets of environmental pollutants, was employed to investigate the potential pathogenic effects of sodium nitrite in AS. A total of 34 sodium nitrite-associated targets were identified from the ChEMBL, CTD, and STITCH databases. Simultaneously, 6,157 AS-related targets were retrieved from the GeneCards and OMIM databases after duplicates were removed. A Venn diagram (Fig 1) revealed 24 overlapping targets, which were subsequently defined as the potential pathogenic targets of sodium nitrite in AS.

## Functional enrichment analysis of sodium nitrite-induced atherosclerosis targets

Functional enrichment analysis was conducted on the 24 overlapping targets. Gene Ontology (GO) terms were selected based on the lowest false discovery rate (FDR) and subsequently visualized (Fig 2A–C). In the Biological Process (BP)

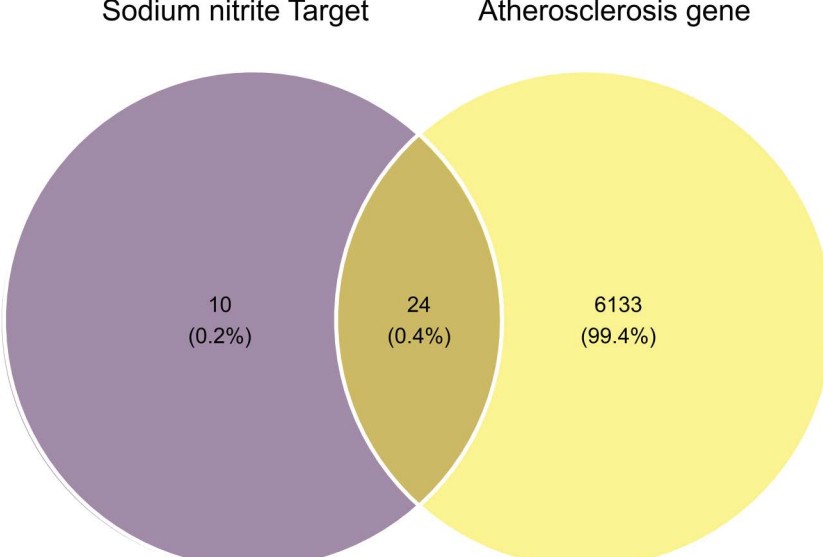

Sodium nitrite Target          Atherosclerosis gene

10
(0.2%)

24
(0.4%)

6133
(99.4%)

**Fig 1. Venn diagram showing the overlap between sodium nitrite-related targets and AS-associated genes.**

category, the targets were primarily enriched in "leukocyte migration," "response to molecule of bacterial origin," and "response to lipopolysaccharide" (Fig 2A). In the Cellular Component (CC) category, the top three terms—"membrane microdomain," "membrane raft," and "neuron spine"—indicated potential involvement in signal transduction (Fig 2B). For the Molecular Function (MF) category, significant enrichment was observed in "organic acid binding," "carboxylic acid binding," and "NADP binding," suggesting roles in intercellular signaling, immune regulation, and inflammatory responses (Fig 2C). Additionally, KEGG pathway enrichment identified associations with "lipid and atherosclerosis," "malaria," and the "AGE–RAGE signaling pathway" (Fig 2D).

### Identification of key genes using machine learning

LASSO and SVM algorithms were employed to screen core genes from the 24 sodium nitrite-related AS targets. Using LASSO logistic regression with three-fold cross-validation, four potential core targets were identified (Fig 3A, B). SVM with 10-fold cross-validation further narrowed the selection to five potential targets (Fig 3C). By integrating the results of both methods with a Venn diagram, four core targets associated with sodium nitrite-induced AS were ultimately identified: IL-1β, PTK2, IL6, and NOS3 (Fig 3D). The identification of these four genes by both LASSO and SVM-RFE algorithms increases confidence in their relevance to sodium nitrite-induced AS. IL-1β and IL6 are well-established pro-inflammatory cytokines central to atherosclerosis, while PTK2 (focal adhesion kinase) plays a role in vascular remodeling and NOS3 regulates endothelial function through nitric oxide production. The convergence of machine learning results on these mechanistically plausible targets supports the validity of our feature selection approach.

### Expression validation and diagnostic performance of the four key targets

The GEO dataset GSE28829 was used to validate the expression of the identified core genes. Compared with normal tissues, NOS3, IL6, and IL-1β were significantly upregulated in AS plaque tissues (Fig 4A, 4B, 4D), whereas PTK2 expression was significantly downregulated (Fig 4C). Receiver operating characteristic (ROC) curves were then generated, and the area under the curve (AUC) values were calculated to assess diagnostic performance. The resulting AUCs were as

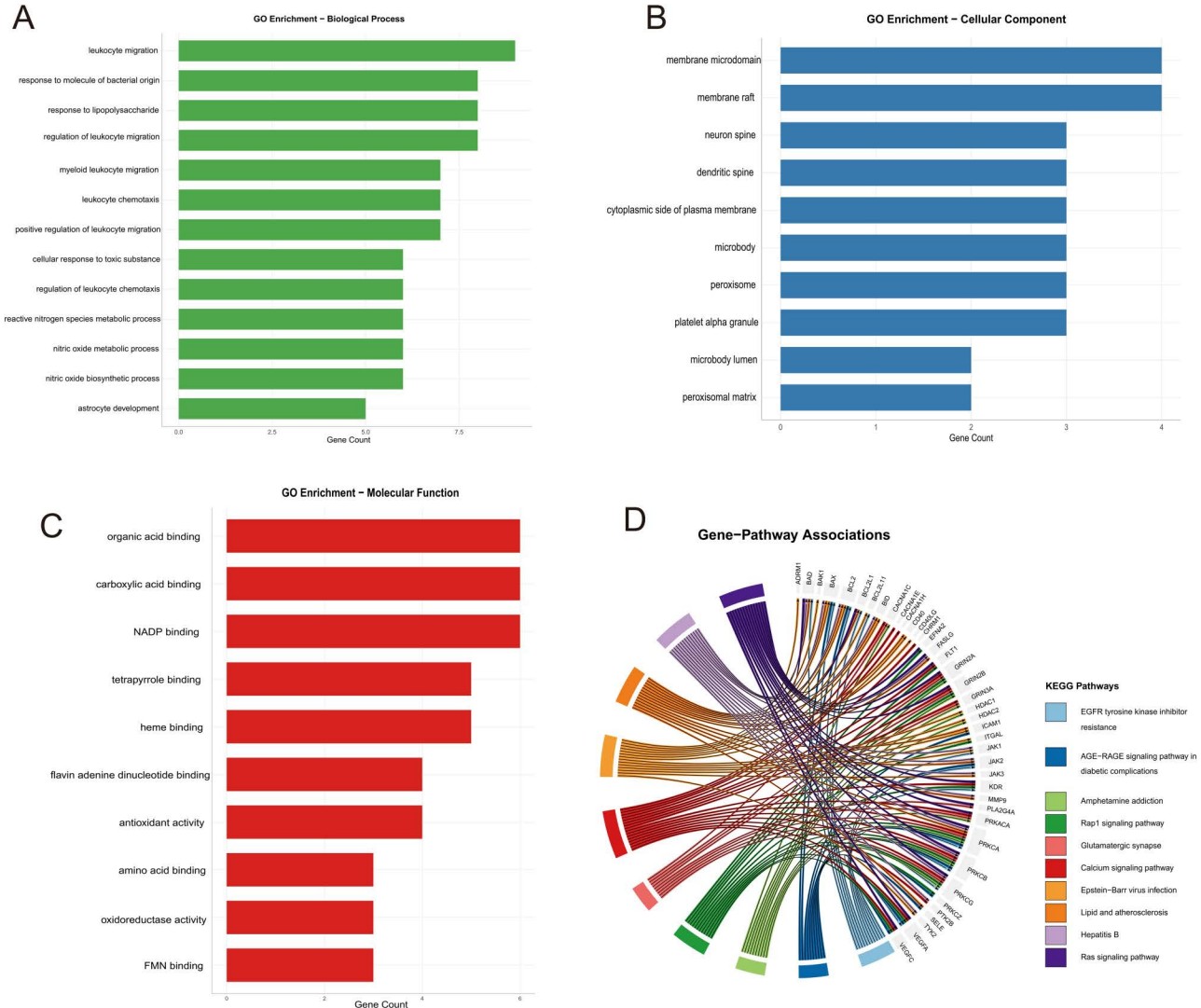

**Fig 2. Gene Ontology (GO) and Kyoto Encyclopedia of Genes and Genomes (KEGG) pathway enrichment analysis of sodium nitrite-induced AS targets.** (A) Biological process (BP). (B) Cellular component (CC). (C) Molecular function (MF). (D) KEGG pathway enrichment, displaying the most significantly enriched pathways associated with the identified targets.

follows: NOS3 (0.975), IL6 (0.810), IL-1β (0.887), and PTK2 (0.715) (Fig 5A–D). Collectively, these findings underscore the potential diagnostic and mechanistic relevance.

To further address the potential risk of overfitting associated with single-cohort validation, we performed external validation using an independent GEO dataset (GSE100927). In this external cohort, IL-1β showed the best-performing and most robust diagnostic discrimination, achieving an AUC of 0.822 (95% CI: 0.7372–0.9069, DeLong). Using the Youden index, the optimal cutoff (7.058918) yielded a sensitivity of 70.4% and a specificity of 84.8%. In comparison, the AUCs of the other three genes in GSE100927 were PTK2 (0.751), IL6 (0.471), and NOS3 (0.618), indicating that IL-1β was the most consistently discriminative marker across datasets(Fig 6A-D). To evaluate the performance of the model, the accuracy,

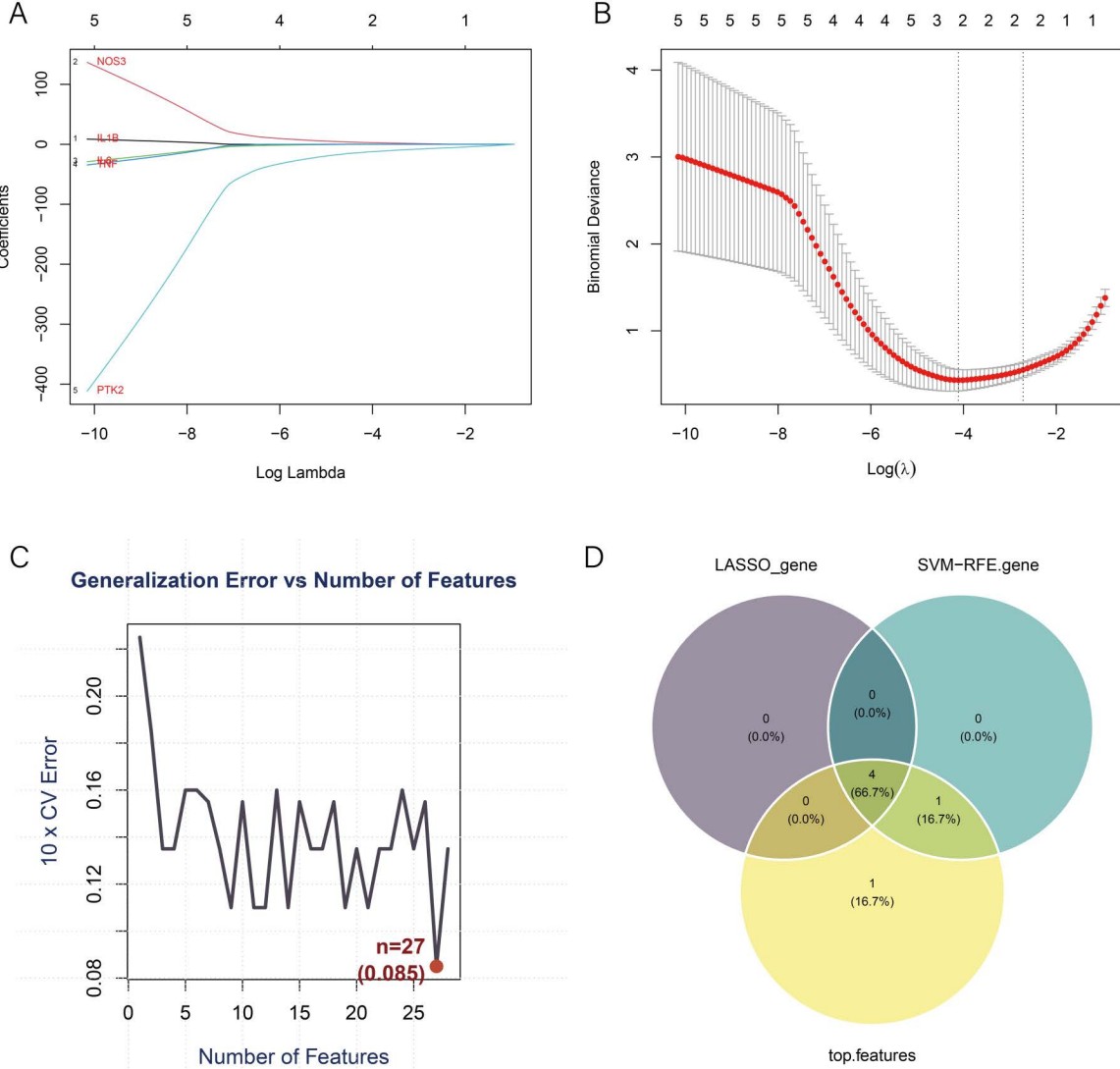

**Fig 3. Screening of core targets of sodium nitrite-induced AS using machine learning algorithms.** (A) LASSO coefficient profile plot showing shrinkage paths of representative variables (e.g., NOS3, IL1B) as the log of the regularization parameter (Log Lambda) increases. (B) Partial likelihood deviance plot, illustrating model deviance at different Log($\lambda$) values. The red dotted line indicates the cross-validation mean, and the gray area represents the confidence interval. (C) Identification of core genes using the SVM–RFE algorithm. The plot shows changes in 10-fold cross-validation error as the number of features increases, with the lowest error occurring when ~27 features are included. (D) Venn diagram showing the intersection of genes identified by LASSO and SVM-RFE, yielding four common core targets of sodium nitrite-induced AS.

recall, and F1 score of the core objective were also calculated. The accuracy of IL-1β is 1, the recall rate is 0.875, and the F1 score is 0.933, showing excellent performance in distinguishing AS related biomarkers. These results emphasize the robustness of the model and support the diagnostic relevance of IL-1 β in atherosclerosis.

## Correlation analysis between core genes and immune cell infiltration in atherosclerosis

To elucidate the immune microenvironment in AS, we employed single-sample Gene Set Enrichment Analysis (ssGSEA) to quantify the relative infiltration levels of 21 immune cell types in AS plaque tissues (n = 27) compared to normal arterial

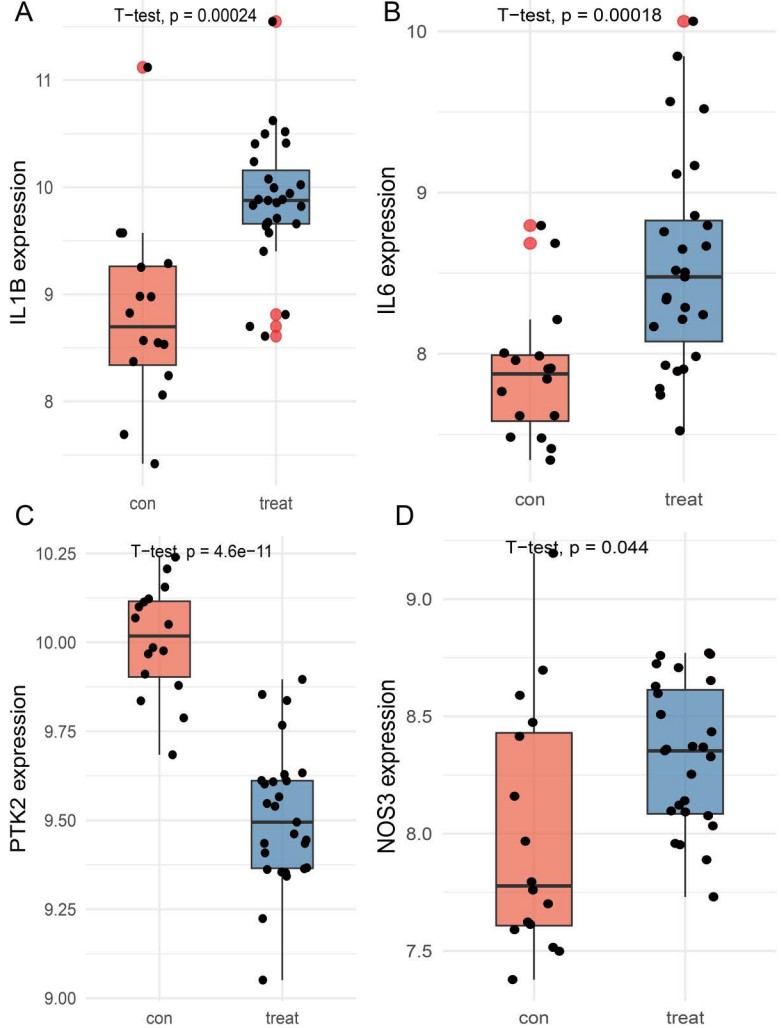

**Fig 4. Expression of the four target genes in normal and AS tissues.** (A) IL-1β. (B) IL6. (C) PTK2. (D) NOS3.

tissues (n = 14). The overall distribution of immune cells is visualized in Fig 7A. Statistical comparison using the Wilcoxon rank-sum test revealed significant differences in the infiltration of specific immune subsets between the two groups (Fig 7C). Crucially, we observed a statistically significant increase in the infiltration of non-polarized (M0) macrophages in AS tissues compared to normal controls (p < 0.001), suggesting their potential role in initiating and sustaining plaque inflammation. In contrast, the infiltration levels of neutrophils, M1 macrophages, M2 macrophages, regulatory T cells (Tregs), and resting NK cells were significantly higher in normal tissues (all p < 0.05). Furthermore, Spearman correlation analysis was performed to assess the interrelationships among different immune cell types (Fig 7B). Finally, to explore the mechanistic links between the core targets and the immune landscape, we evaluated the correlations between the expression of the four core genes (IL-1β, PTK2, IL6, NOS3) and the infiltration levels of these differentially abundant immune cells.

### Molecular docking analysis of sodium nitrite with atherosclerosis targets

Molecular docking was performed to evaluate interactions between sodium nitrite and the four core proteins. Sodium nitrite showed moderate to strong binding affinities with IL-1β, PTK2, IL6, and NOS3, with binding energies of –6.9, –6.7, –6.3, and

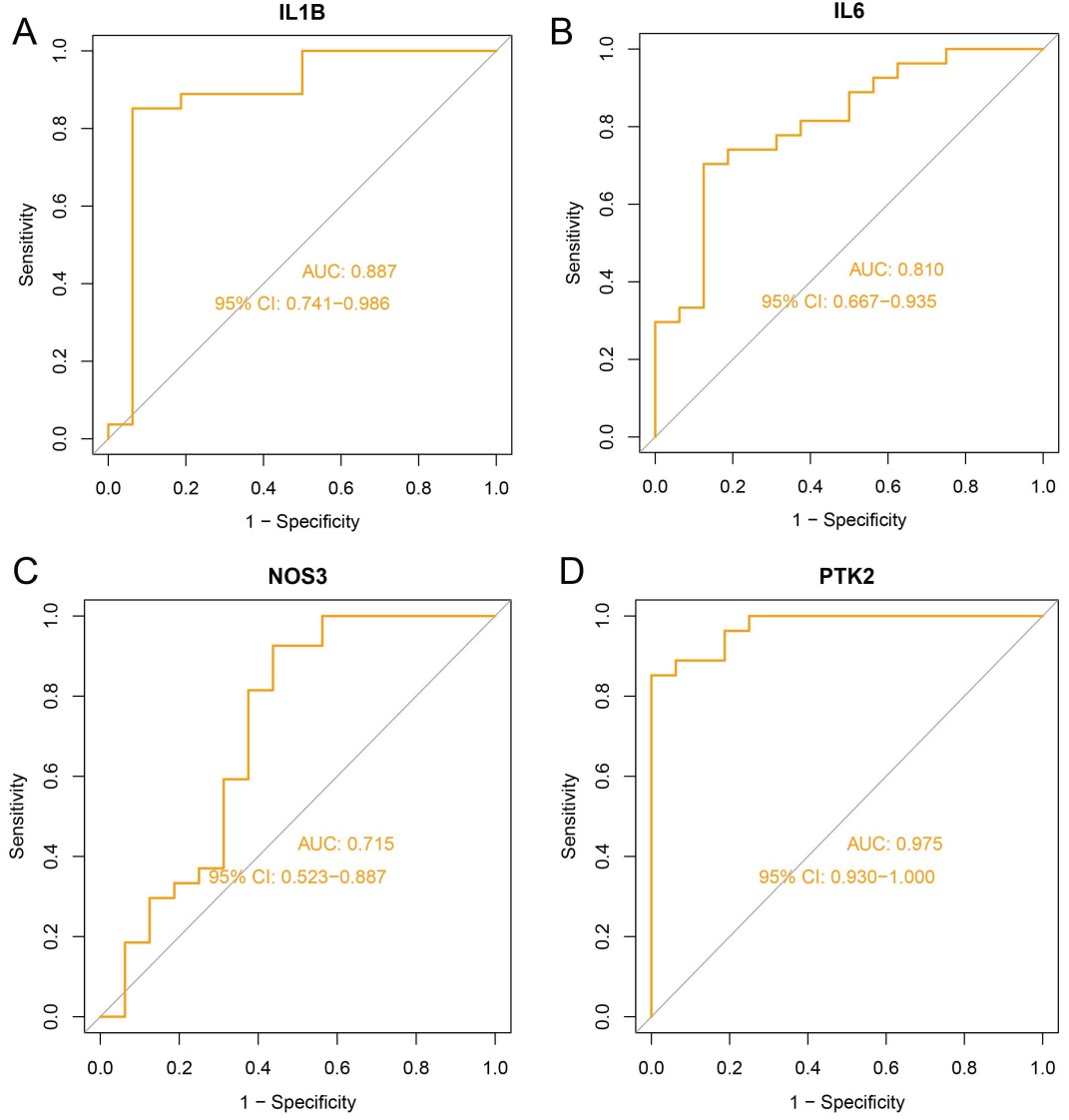

**Fig 5. Receiver operating characteristic (ROC) curves evaluating the diagnostic performance of the four core targets in distinguishing normal from AS samples.** (A) IL-1β. (B) IL6. (C) NOS3. (D) PTK2.

−6.8 kcal/mol, respectively (Fig 8A–D). Binding energies lower than −5.0 kcal/mol indicated spontaneous binding potential. To elucidate the specific molecular binding mode, a 2D ligand protein interaction diagram was generated. Analysis shows that sodium nitrite mainly binds to the core target through hydrogen bonding, salt bridging, and hydrophobic interactions. Among them, the interaction with IL-1β (binding energy −6.9 kcal/mol) is the most significant, forming a stable salt bridge with the key residue Arg A: 4 and hydrogen bonding with Gln A: 48, which provides a structural explanation for its binding affinity. The 2D images of PTK2, IL6, and NOS3 also reveal similar specific interaction networks. In addition, the docking results of the positive control MLN4924 with IL-1β showed a broader interaction, verifying the reliability of the docking method, while highlighting the different and more focused binding characteristics of sodium nitrite. These structural details collectively reinforce the evidence that sodium nitrite can directly act on these core targets. Among these, the lowest binding energy was observed for IL-1β (−6.9 kcal/mol), suggesting that IL-1β is the strongest sodium nitrite-binding target. For comparative

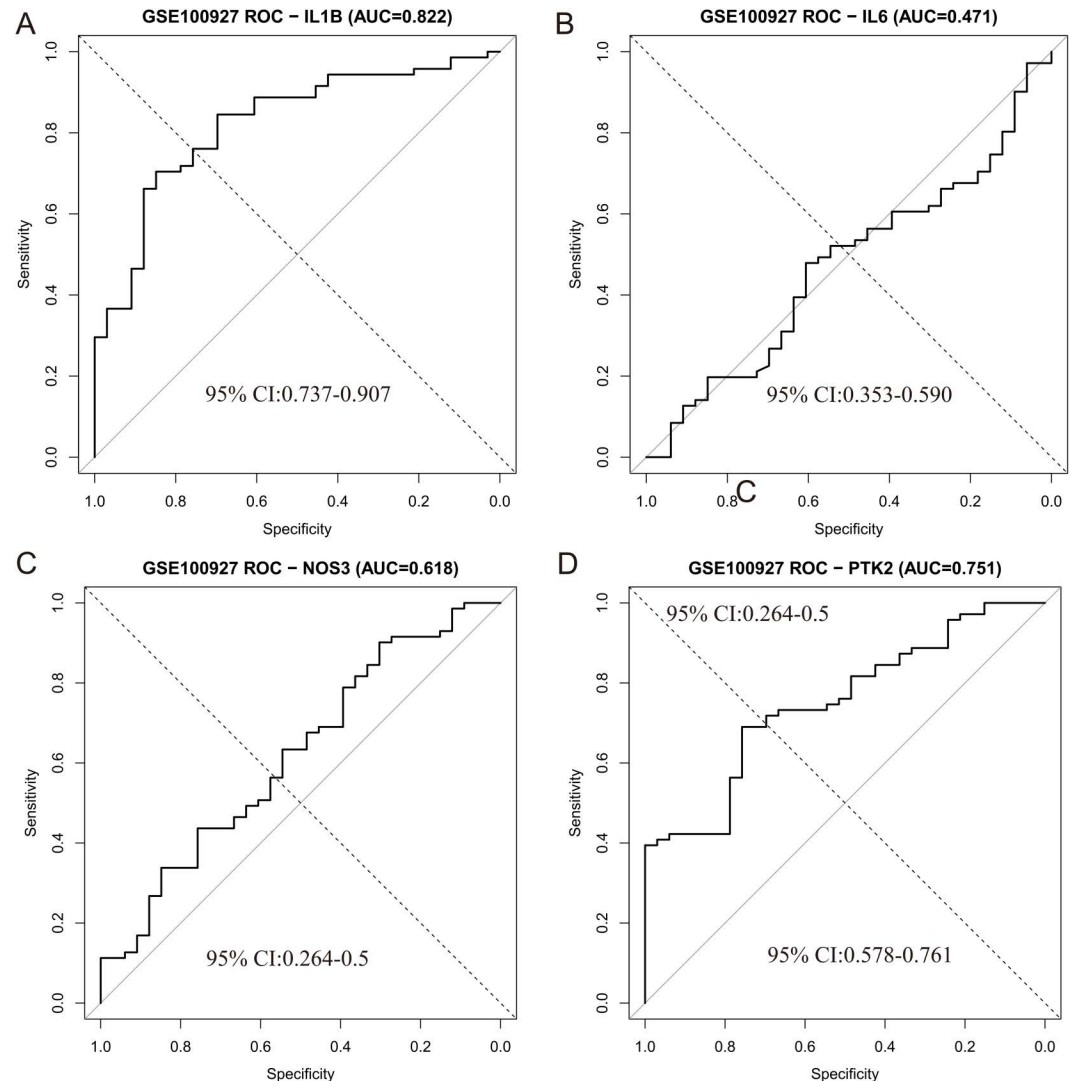

**Fig 6. External validation of diagnostic performance for the four key genes in GSE100927.** (A) IL-1β. (B) IL6. (C) NOS3. (D) PTK2.

purposes, we also docked MLN4924, a known inhibitor of the IL-1β signaling pathway, to IL-1β. The significantly stronger binding energy of MLN4924 (−8.9 kcal/mol) provides a reference, indicating that the binding affinity of sodium nitrite for IL-1β is moderate to strong. These binding energies, all below the −5.0 kcal/mol threshold for spontaneous binding, suggest that sodium nitrite has the structural potential to directly interact with all four core targets. The strongest affinity observed for IL-1β (−6.9 kcal/mol) is particularly noteworthy, as it aligns with the central role of IL-1β in our subsequent MD simulations and immune infiltration analyses. While docking scores alone cannot confirm biological activity, they provide a basis for hypothesizing direct molecular interactions that warrant experimental validation.

## Molecular dynamics simulation results

To further examine the stability of protein--ligand interactions, MD simulations were conducted for the IL-1β--sodium nitrite complex. Multiple independent metrics were analyzed to comprehensively assess the structural stability and binding

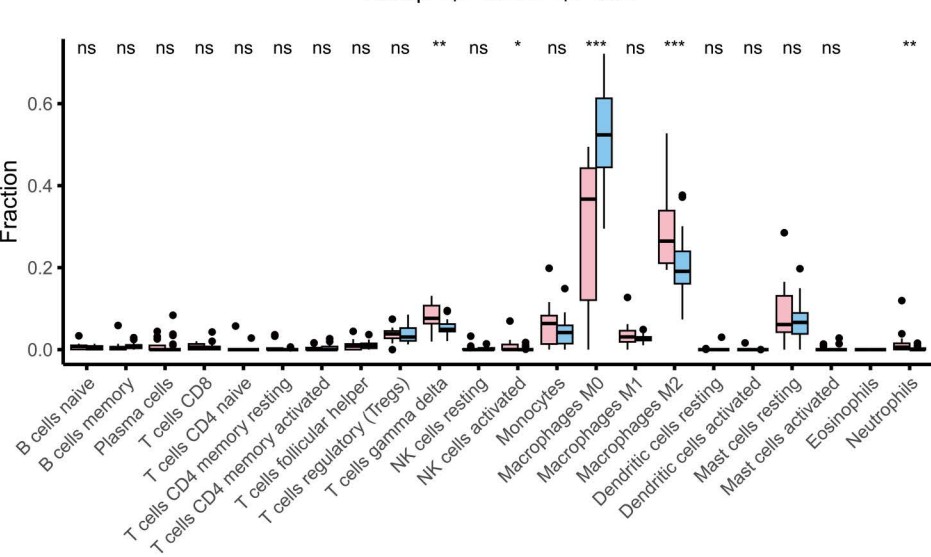

**Fig 7. Correlation analysis between target genes and immune infiltration scores.** (A) Bar plot of relative immune cell proportions. (B) Heatmap of immune cell abundance. (C) Boxplots of significantly different immune cell populations.

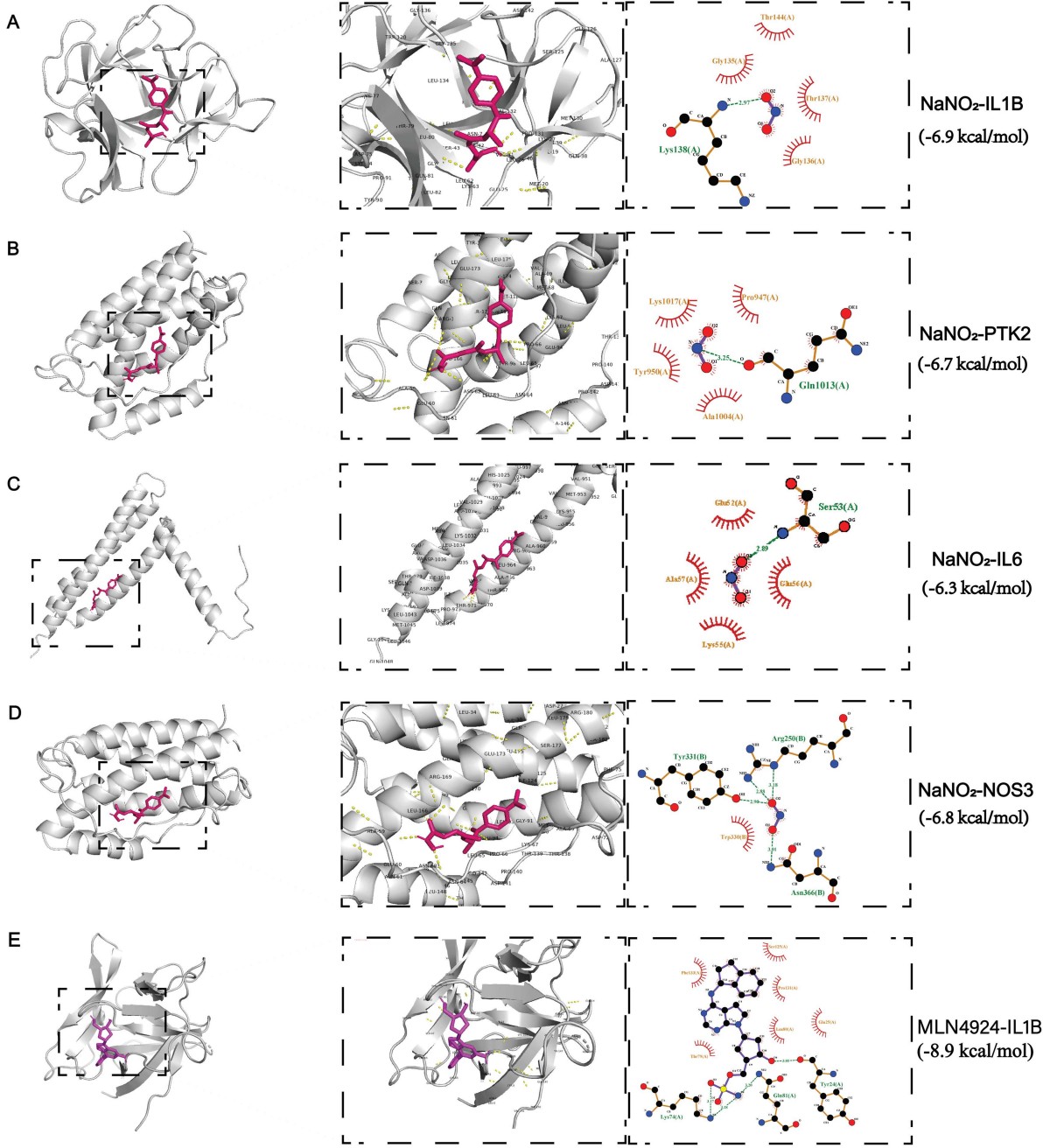

**Fig 8. Molecular docking results showing the binding affinities of sodium nitrite with core target proteins.** (A) IL-1β. (B) PTK2. (C) IL6. (D) NOS3. (E)Molecular docking between MLN4924 and IL-1β.

persistence of the complex. Protein backbone RMSD analysis (Fig 9A) revealed a stable trajectory throughout the 100 ns simulation. Following an initial equilibration phase during the first 20 ns, the RMSD stabilized and remained within a narrow range (approximately 0.2–0.3 nm) with no major fluctuations, indicating that the protein maintained its structural integrity. Ligand RMSD analysis (Fig 9B) showed moderate fluctuations during the first 40 ns, which correspond to the

ligand exploring its binding pose within the pocket—a common phenomenon in protein–ligand simulations. After this initial sampling period, the ligand RMSD stabilized and remained confined for the latter 60 ns, indicating the establishment of a stable binding mode. Critically, the radius of gyration (Rg) remained largely constant throughout the simulation (Fig 9D), demonstrating that the protein retained its compact, globular structure and did not undergo global unfolding. This is a key indicator that the observed backbone RMSD changes reflect localized conformational adjustments rather than overall destabilization. The solvent-accessible surface area (SASA) displayed only minor fluctuations (Fig 9E), further supporting the maintenance of structural integrity. Root mean square fluctuation (RMSF) analysis (Fig 9C) showed expected flexibility

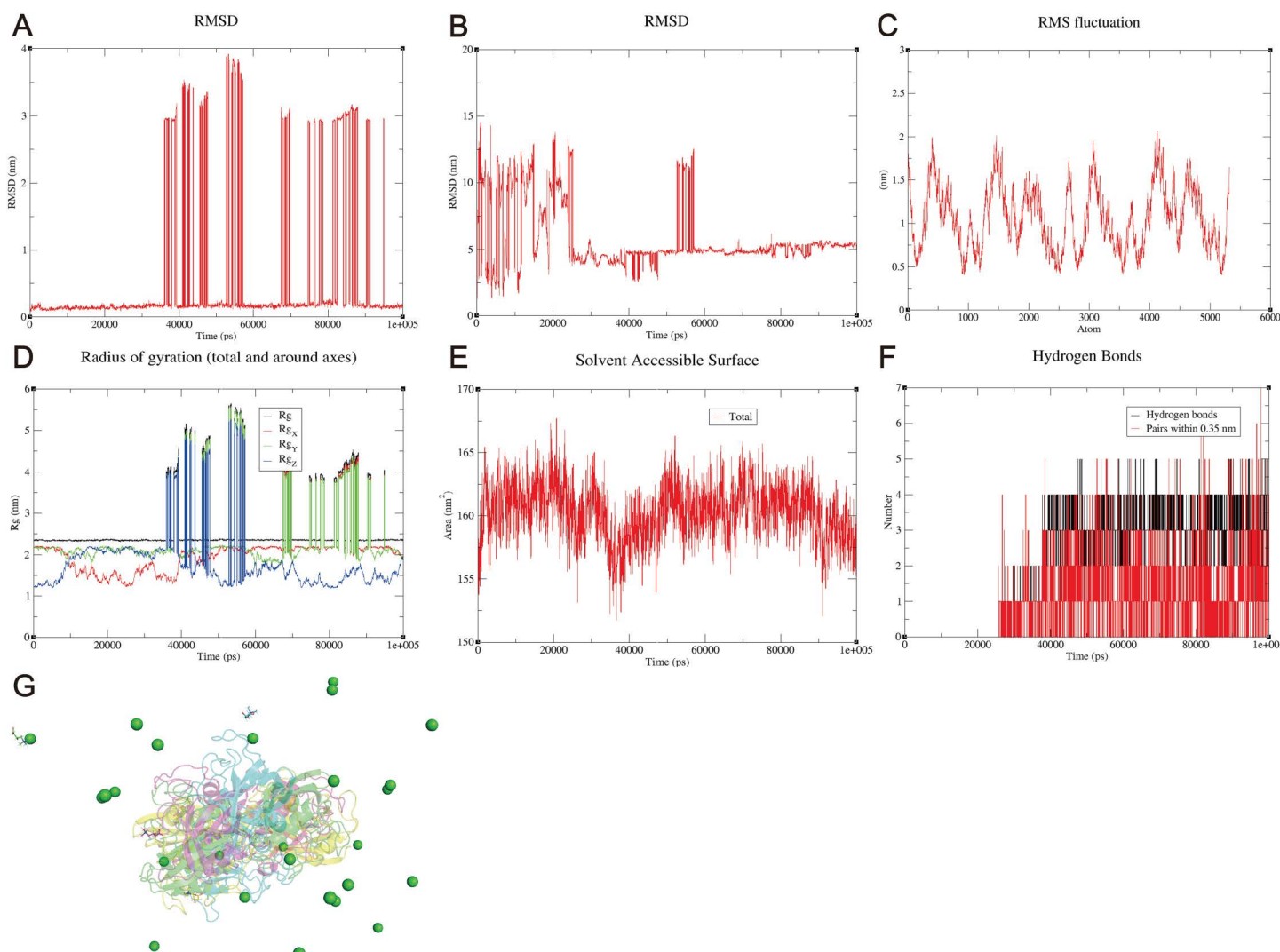

**Fig 9. Molecular dynamics (MD) simulation results demonstrating the structural stability and interaction persistence of the sodium nitrite–IL-1β complex.** (A) RMSD plot of the IL-1β protein backbone. (B) RMSD plot of the sodium nitrite ligand. (C) RMSF profile of residues in the sodium nitrite–IL-1β complex. (D) Radius of gyration (Rg) of the sodium nitrite–IL-1β complex (total and per axis). (E) Solvent-accessible surface area (SASA) of the sodium nitrite–IL-1β complex. (F) Hydrogen-bond occupancy/contact persistence between sodium nitrite and IL-1β during the simulation (cutoff as defined in the analysis). (G) Representative snapshots of the ligand binding pose at 0 ns (blue), 25 ns (cyan), 50 ns (green), and 75 ns (red). The protein is shown as a transparent gray cartoon. Surrounding residues within 4 Å are shown as yellow sticks.

in loop regions and termini, while residues surrounding the binding pocket exhibited lower fluctuations, supporting a stable interaction interface. Most importantly, hydrogen bond analysis (Fig 9F) revealed persistent interactions between sodium nitrite and IL-1β throughout the simulation, confirming that the ligand remained associated with the binding pocket despite positional fluctuations [15]. Representative snapshots of the sodium nitrite binding pose at 0, 25, 50, and 75 ns (Fig 9G) visually confirm the stable binding mode inferred from the quantitative analyses, showing minimal positional drift and consistent orientation of the ligand throughout the trajectory. Collectively, these multiple lines of evidence—stable Rg, persistent hydrogen bonds, confined ligand RMSD after initial sampling, and localized RMSF patterns—demonstrate that the RMSD variations observed reflect expected dynamic conformational adjustments rather than unstable binding. The system reached a stable equilibrium, validating the overall stability of the sodium nitrite–IL-1β complex.

### Free energy landscape analysis

The Gibbs free energy landscape of the IL-1β–sodium nitrite binding process was constructed using principal component analysis (PCA)-based MD trajectories. In the conformational space defined by PC1 and PC2, two distinct free energy minima were observed, corresponding to the unbound and bound conformational states, separated by an energy barrier of 12.5–17.5 kJ/mol (Fig 10A). This barrier represents the activation energy required for binding and is characteristic of biologically specific interactions. A three-dimensional surface plot revealed a deep, narrow free energy well (~5 kJ/mol) in the high PC1/PC2 region, confirming the high stability of the sodium nitrite–IL-1β complex. Moreover, the ~15 kJ/mol energy difference between the bound-state well and the transition-state peak (~20 kJ/mol) suggests a binding and dissociation process that is feasible within experimentally observable timescales, thereby ensuring both specificity and kinetic plausibility (Fig 10B). The presence of two distinct energy minima corresponding to unbound and bound states, separated by a moderate energy barrier (12.5–17.5 kJ/mol), indicates that sodium nitrite binding to IL-1β is both thermodynamically favorable and kinetically accessible. The deep, narrow energy well in the bound state region (~5 kJ/mol) further supports the high stability of the complex observed in the RMSD and hydrogen bond analyses. This energy landscape is characteristic of specific, biologically relevant protein–ligand interactions rather than non-specific, transient binding.

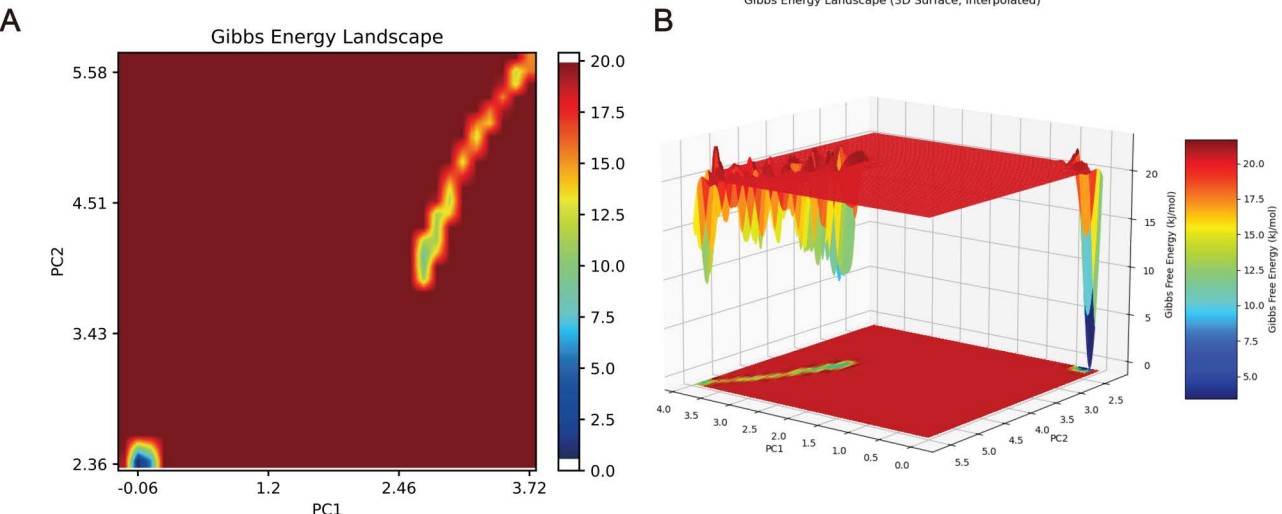

**Fig 10. Gibbs free energy landscape of IL-1β binding with sodium nitrite.** (A) 2D projection of the free energy landscape. (B) 3D free energy surface plot of the IL-1β–sodium nitrite interaction.

## Construction of an adverse outcome pathway

A novel Adverse Outcome Pathway (AOP) framework was constructed [16]. Within this framework, the expression and activity of IL-1β, PTK2, IL6, and NOS3 may be modulated by sodium nitrite, triggering downstream effects mediated through the TNF, AGE–RAGE, IL-17, and chemokine signaling pathways. These cascades disrupt lipid metabolism and promote chronic inflammation, ultimately driving the progression of AS (Fig 11). Collectively, this integrative AOP framework provides a theoretical basis for elucidating the mechanisms by which sodium nitrite exacerbates atherosclerosis.

## Discussion

Sodium nitrite is most commonly present in processed meats and pickled vegetables, where it is legally used as a food additive to retain color, preserve freshness, and enhance flavor [17]. Its association with AS may involve multiple upstream events, including (i) endogenous nitrosation/nitrosative stress and (ii) a putative direct interaction with inflammatory mediators (e.g., IL-1β) suggested by our in silico analyses. These compounds are produced when sodium nitrite reacts with protein degradation products in the acidic gastric environment [18]. These nitrosamines can trigger chronic inflammation, which indirectly damages the vascular endothelium and promotes plaque formation. Moreover, foods rich in sodium nitrite are often high in salt, and the combined effects of excessive salt intake and sodium nitrite further aggravate hypertension and endothelial dysfunction. Together, these factors contribute to the progression of atherosclerosis [19]. Given the complexity of the effects of sodium nitrite, several epidemiological studies have explored the potential role of dietary nitrite/nitrate in cardiovascular health. A large-scale cohort study in 2025 found that excessive intake of plant nitrates is associated with a reduced risk of cardiovascular disease and mortality, possibly due to the beneficial effects of NO production on vascular function and health [20]. However, the intake of nitrites from processed meats—where levels are much higher—has been associated with an increased risk of cardiovascular diseases. The presence of high levels of nitrites in processed foods is often linked to the formation of harmful nitrosamines under acidic conditions in the stomach. These compounds can lead to chronic inflammation and endothelial dysfunction, which contribute to the development of atherosclerosis [21].

International safety standards have been established for sodium nitrite exposure. The World Health Organization (WHO) has established a guideline for nitrite in drinking water, with a recommended limit of 3 mg/L (3000 µg/L) for nitrite, specifically in the context of infant health. For food, nitrite concentrations in processed meats are regulated under national or regional standards, such as those established by the U.S. Food and Drug Administration (FDA) and the European Food Safety Authority (EFSA), which set limits that vary between 100–200 mg/kg, depending on the product type [22,23]. However, these standards are primarily focused on short-term exposure and acute toxicity risks, while the health effects of long-term low-dose exposure remain insufficiently studied. Based on the findings of this study, sodium nitrite exacerbates the occurrence of AS through mechanisms such as oxidative stress and endothelial damage, but the specific relationship between low-dose, long-term exposure and AS still lacks a clear threshold definition. Although the existing standards provide protection for acute

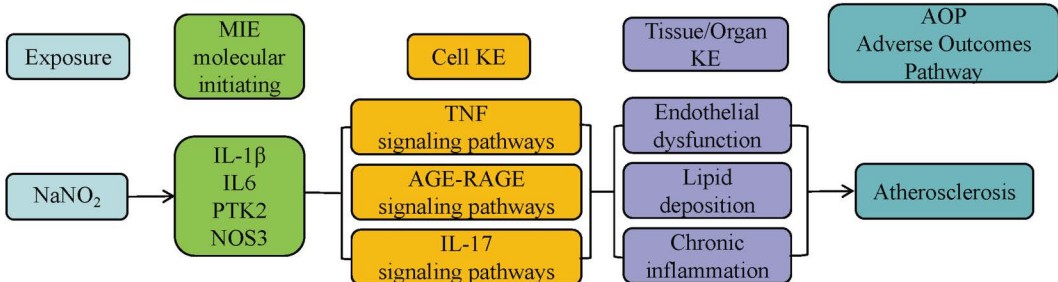

**Fig 11. Proposed adverse outcome pathway (AOP) framework for sodium nitrite-induced atherosclerosis.**

exposure, further research is needed for risk assessment regarding chronic exposure, especially considering individual differences, such as smokers or individuals with hypertension, who may be more sensitive to sodium nitrite exposure [24].

In this study, we identified IL-1β, PTK2, IL-6, and NOS3 as core targets associated with AS progression. These four targets play important roles in regulating inflammation, endothelial dysfunction, and vascular remodeling. IL-1β is a key proinflammatory cytokine in the occurrence and spread of atherosclerosis plaque inflammation. Elevated levels of IL-1β are associated with increased endothelial dysfunction, immune cell infiltration, and plaque instability, all of which contribute to the progression of atherosclerosis. Importantly, the CANTOS trial showed that targeting IL-1β with canakinumab can significantly reduce recurrent cardiovascular events, emphasizing its clinical relevance in atherosclerosis [25,26]. These findings are supported by extensive experimental evidence, positioning IL-1β as a key mediator of vascular inflammation and plaque instability, making it a promising therapeutic target for reducing AS related cardiovascular risk. PTK2 plays a crucial role in vascular remodeling, regulating cell adhesion, migration, and signal transduction, particularly in vascular smooth muscle cells and endothelial function. PTK2 (Focal Adhesion Kinase) activation can enhance vascular inflammation, promote immune cell recruitment, and facilitate plaque progression. This makes PTK2 an important target for stabilizing plaques and reducing the risk of plaque rupture [27]. IL-6 is another pro-inflammatory cytokine involved in immune cell activation, acute phase response, and plaque formation. Elevated IL-6 levels are associated with vascular inflammation, plaque instability, and poor cardiovascular prognosis. It interacts with other inflammatory mediators such as TNF-α, amplifying the inflammatory response of the vascular system. Targeting IL-6 can help alleviate the inflammation that drives the progression of atherosclerosis [28]. NOS3 is responsible for the production of NO. Nitric oxide is a molecule that regulates vasodilation, antithrombotic effects, and endothelial protection. The decrease in NOS3 activity leads to endothelial dysfunction, which is a hallmark of atherosclerosis. NOS3 dysfunction is associated with vascular stiffness, impaired vasodilation, and plaque instability, making it a potential therapeutic target for restoring endothelial function and slowing down the progression of atherosclerosis [29].

In conclusion, IL-1β, PTK2, IL-6 and NOS3 play different roles in vascular inflammation, immune cell recruitment and plaque instability, all of which are the core of atherosclerosis pathogenesis. Given their role in AS, these targets are valuable candidates for therapeutic interventions.

To further reduce the potential risk of overfitting from single-cohort validation, we performed external validation using an independent GSE100927. In this external cohort, IL-1β showed the best-performing and most robust diagnostic discrimination, achieving an AUC of 0.822 (95% CI: 0.7372–0.9069, DeLong). Using the Youden index, the optimal cutoff (7.058918) yielded a sensitivity of 70.4% and a specificity of 84.8%. Notably, GSE100927 represents an AS cohort without sodium nitrite exposure information; therefore, this external validation supports the reproducibility of IL-1β as an AS-related diagnostic marker rather than providing direct evidence that sodium nitrite regulates IL-1β in vivo. Future in vitro and in vivo studies using sodium nitrite exposure models are required to establish causality and clarify whether these candidate genes are directly affected by sodium nitrite or reflect downstream pathological processes in AS.

This external validation was intended to strengthen the robustness of the diagnostic findings and does not change the hypothesis-generating nature of the upstream target collection.Adding additional machine learning metrics, including accuracy, recall, and F1 score, can further understand the reliability of our predictions. Although AUC provides a broad understanding of diagnostic performance, the high accuracy (1), recall rate (0.875), and F1 score (0.933) values indicate that our model is suitable for identifying potential biomarkers associated with AS. However, it is worth noting that these results are based on computational predictions and require further experimental validation to confirm the clinical relevance and accuracy of IL-1β as a therapeutic target for AS. The performance metrics highlight the robustness of the model in distinguishing key biomarkers, but additional in vitro and in vivo studies are crucial for establishing causal relationships and deepening our understanding.

Sodium nitrite exerts a dose-dependent dual effect on vascular health. In experimental settings, sodium nitrite may show a biphasic profile. In experimental settings, sodium nitrite may show a biphasic profile, with low doses (1−20 mM) acting through the NO pathway to promote vasodilation, reduce blood pressure, and prevent LDL oxidation. These effects

help mitigate atherosclerosis progression. However, higher doses lead to the production of RNS, which induce oxidative stress and inflammation, accelerating atherosclerosis. The exact dose-response relationship and time-dependent effects require further investigation. We hypothesize that sodium nitrite may activate IL-1β signaling, a central pro-inflammatory mediator in atherosclerosis. Previous studies have validated IL-1β's role in initiating and maintaining vascular inflammation, as well as in promoting plaque instability. High doses of sodium nitrite lead to the production of RNS, such as peroxynitrite, which activate IL-1β and increase the levels of pro-inflammatory cytokines, resulting in endothelial injury and plaque destabilization. This inflammatory cascade highlights the critical role of IL-1β in atherosclerosis, suggesting that sodium nitrite exposure could potentiate IL-1β signaling, contributing to disease progression.However, the precise causal relationship between sodium nitrite exposure and IL-1β modulation remains to be validated experimentally. Future in vitro and in vivo studies using sodium nitrite exposure models will be crucial to confirm this hypothesis.

IL-1β and TNF-α have been extensively validated as central mediators of inflammation in AS. IL-1β triggers immune cell infiltration, including macrophages, and activates downstream inflammatory signaling pathways such as IL-17 and AGE-RAGE [30,31]. This process may be exacerbated by sodium nitrite, which could potentiate IL-1β-mediated immune responses in AS.However, the relationship between sodium nitrite exposure and IL-1β modulation remains hypothesis-generating, and further in vitro and in vivo validation is needed to confirm causality and better understand how sodium nitrite interacts with IL-1β in vascular tissue.

This bidirectional effect of sodium nitrite highlights the importance of exposure dose and duration in its potential to either protect or exacerbate atherosclerosis. Further studies are needed to determine the exact dose-response relationship and its implications for vascular health.

This study found that sodium nitrite exhibited the strongest binding affinity to IL-1β (–6.9 kcal/mol) and maintained a highly stable binding conformation in molecular dynamics simulations. The 2D interaction diagram further clarified that this high-affinity binding is mediated by a stable salt bridge with the key residue Arg A:4 and a hydrogen bond with Gln A:48 within the IL-1β binding pocket, providing structural rationale for the predicted interaction. These results suggest that sodium nitrite may interact with IL-1β and potentially influence its stability,receptor interaction, thereby amplifying local inflammatory cascades and accelerating both plaque formation and destabilization [32]. Downregulation of NOS3 often results in reduced nitric oxide production, thereby impairing vasodilation and antithrombotic functions. At the same time, immune infiltration analysis revealed a pronounced enrichment of M0 macrophages within atherosclerotic plaques, accompanied by decreased numbers of immunosuppressive populations such as Tregs. These findings suggest that sodium nitrite–related target genes are closely associated with immune microenvironment remodeling in AS, implying a potential role in mediating toxicity-related inflammatory pathways; however, causality requires validation in sodium nitrite exposure models.Such an immune microenvironment aligns with mechanisms of local immune dysregulation in atherosclerosis recently uncovered by high-throughput single-cell sequencing studies [33].

Within the AOP framework proposed here, sodium nitrite exposure can be conceptually linked to AS through a stepwise causal cascade. As a putative molecular initiating event (MIE), our docking and MD results suggest a plausible interaction between sodium nitrite and IL-1β, which may affect IL-1β availability or downstream signaling potential, although the functional directionality requires biochemical and cellular validation. A subsequent key event (KE) is the amplification of vascular inflammatory signaling, characterized by elevated IL-1β–centered cytokine networks (e.g., TNF/IL-6) and activation of downstream pathways such as IL-17 and AGE–RAGE. These inflammatory KEs are expected to promote endothelial activation and chemokine-driven recruitment of monocytes/macrophages, consistent with our immune infiltration results showing increased macrophage-related signatures in AS plaques. Sustained immune cell infiltration and inflammatory signaling then represent later KEs that facilitate lipid dysregulation, foam cell formation, and plaque growth/instability, culminating in the adverse outcome (AO) of AS progression. Notably, prior studies have established IL-1β as a causal mediator in atherogenesis and plaque destabilization [34], and our findings extend this literature by proposing sodium nitrite as an upstream stressor that may converge on the IL-1β–inflammation axis in an AOP-consistent manner.

It is noteworthy that at low exposure levels, sodium nitrite may exert vasoprotective effects through the NO pathway, improving endothelial relaxation and inhibiting platelet aggregation. However, under high-dose or long-term exposure, its predominant effects are pro-inflammatory and pro-oxidative. This bidirectional effect underscores the need for future studies to incorporate exposure dose, time window, and individual susceptibility to refine the assessment of sodium nitrite cardiovascular risk thresholds. Therefore, to protect cardiovascular health, it is advisable to limit the consumption of foods rich in sodium nitrite while increasing the intake of fresh fruits and vegetables. Vitamin C, known for its antioxidant properties, can potentially inhibit nitrosamine formation, thus mitigating associated health risks [35].

## Limitations and future perspectives

Although this study integrates multi-dimensional bioinformatics and molecular simulation approaches, it still lacks direct validation through in vitro cell experiments and in vivo animal models. Moreover, several key events within the constructed AOP framework are primarily inferred from existing literature, and their causal relationships need further clarification through time-series experiments.

Future research should therefore prioritize the following directions:

(1) Employing animal models to dynamically trace the causal relationship between sodium nitrite exposure and AS progression.

(2) Investigating the bidirectional effects of sodium nitrite under different exposure doses.

(3) Evaluating the therapeutic potential of targeting inflammatory cytokines, such as IL-1β and IL-6, in mitigating sodium nitrite-induced toxic effects.

(4) The 24 overlapping genes were derived from public database annotations and should be considered hypothesis-generating putative sodium nitrite-AS-related targets rather than confirmed direct binding targets, because the overlap may reflect indirect downstream effects, and thus requires future in vitro/in vivo sodium nitrite exposure studies for validation.

(5) Regarding the machine learning approach, this study employed a combination of LASSO regression and SVM-RFE. While these algorithms were selected based on their methodological strengths (LASSO for regularization and feature selection, SVM-RFE for robust feature ranking in linearly separable data), we acknowledge that systematic comparison with other classifiers—such as random forest, XGBoost, or neural networks—was not performed. This represents a limitation of the present study. Future research should incorporate a broader range of machine learning algorithms for comparative evaluation to further validate and optimize the feature selection pipeline.

Therefore, future research should focus on the impact of low-dose long-term exposure on cardiovascular health, define the safety thresholds for sodium nitrite exposure, and explore intervention strategies for diseases related to sodium nitrite exposure.

## Conclusions

This study systematically investigated, using computational approaches, potential mechanisms by which sodium nitrite may contribute to atherosclerosis. By integrating network toxicology, machine learning, molecular docking, and molecular dynamics simulations, we identified IL-1β, IL6, PTK2, and NOS3 as candidate targets, with stable in silico binding of sodium nitrite to IL-1β. The immune-infiltration patterns and the proposed AOP framework provide hypothesis-generating mechanistic insights that warrant validation in sodium nitrite exposure models. Collectively, our findings provide in silico evidence and a hypothesis-generating framework for assessing the potential cardiovascular risk of sodium nitrite exposure, highlighting the need for future experimental validation, especially under long-term low-dose exposure scenarios.

## Supporting information

**S1 File. S1Underlying data supporting the findings of this study.**
(XLSX)

## Acknowledgments

The authors sincerely thank Hunan University of Chinese Medicine for providing support and necessary facilities for this research.

## Author contributions

**Conceptualization:** YaNan Bai, WeiXiong Jian.

**Data curation:** YunFeng Yu.

**Funding acquisition:** WeiXiong Jian.

**Investigation:** HaoBo Yang, YongHui Zhang, YaNan Bai, YaRu Shi.

**Resources:** HaoBo Yang.

**Supervision:** WeiXiong Jian.

**Visualization:** HaoBo Yang.

**Writing – original draft:** HaoBo Yang.

**Writing – review & editing:** YunFeng Yu, YongHui Zhang, YaRu Shi, WeiXiong Jian.

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
