## [Decision Letter · Decision Letter 0]

9 Dec 2025

PONE-D-25-58627Sodium nitrite promotes atherosclerosis via IL-1β: Network toxicology and machine learning insightsPLOS One

Dear Dr. Yang,

Thank you for submitting your manuscript to PLOS ONE. After careful consideration, we feel that it has merit but does not fully meet PLOS ONE’s publication criteria as it currently stands. Therefore, we invite you to submit a revised version of the manuscript that addresses the points raised during the review process.

If applicable, we recommend that you deposit your laboratory protocols in protocols.io to enhance the reproducibility of your results. Protocols.io assigns your protocol its own identifier (DOI) so that it can be cited independently in the future. For instructions see: https://journals.plos.org/plosone/s/submission-guidelines#loc-laboratory-protocols. Additionally, PLOS ONE offers an option for publishing peer-reviewed Lab Protocol articles, which describe protocols hosted on protocols.io. Read more information on sharing protocols at . Additionally, PLOS ONE offers an option for publishing peer-reviewed Lab Protocol articles, which describe protocols hosted on protocols.io. Read more information on sharing protocols at https://plos.org/protocols?utm_medium=editorial-email&utm_source=authorletters&utm_campaign=protocols..

We look forward to receiving your revised manuscript.

Kind regards,

Wenxing Li

Academic Editor

PLOS One

“The authors declare that the research, writing and publication of this paper were supported by the National Natural Science Foundation of China (82374334) , Natural Science Foundation of Hunan Province (2024JJ9466)and Hunan Province Graduate Research Innovation Project (CX20251169).”

“This work was supported by the following:

National Natural Science Foundation of China (Grant No. 82374334). URL: https://www.nsfc.gov.cn/

Natural Science Foundation of Hunan Province (Grant No. 2024JJ9466). URL: http://kjt.hunan.gov.cn/

Graduate Innovation Project of Hunan Province (Grant No. CX20251169). URL: http://jyt.hunan.gov.cn/

5. Please amend either the title on the online submission form (via Edit Submission) or the title in the manuscript so that they are identical.

Reviewers' comments:

Reviewer's Responses to Questions

**Comments to the Author**

1. Is the manuscript technically sound, and do the data support the conclusions?

Reviewer #1: Partly

Reviewer #2: Partly

2. Has the statistical analysis been performed appropriately and rigorously? 

Reviewer #1: Yes

Reviewer #2: Yes

3. Have the authors made all data underlying the findings in their manuscript fully available?

Reviewer #1: Yes

Reviewer #2: Yes

4. Is the manuscript presented in an intelligible fashion and written in standard English?

Reviewer #1: Yes

Reviewer #2: Yes

5. Review Comments to the Author

Reviewer #1: Dear Editor and Authors,

Thank you for inviting me to review a manuscript for this esteemed journal.

My review comments on the requested manuscript (PONE-D-25-58627) are as follows.

# Overall Comments

This manuscript employs network toxicology and multifaceted bioinformatics methodologies to elucidate the mechanisms by which sodium nitrite contributes to atherosclerosis development.

This represents a valuable approach for unravelling complex toxicological mechanisms, and the scientific merit of both the subject matter and the findings is considerable.

However, to fully meet PLOS ONE's publication standards, methodological rigour should be enhanced in the following aspects.

# suggestions

1. The manuscript's methodology describes that the authors collected targets for sodium nitrite via ChEMBL, STITCH, and CTD, and AS-related targets via OMIM and GeneCards.

However, the GeneCards database returns thousands of genes based on their relevance score. This manuscript merely states that ‘duplicate entries were removed’, without specifying a concrete cut-off value for the Relevance Score.

Simply including all genes risks incorporating false positive targets, which could undermine the reliability of the results.

Therefore, please specify the exact filtering criteria applied to the GeneCards database and provide the rationale for selecting these criteria.

Furthermore, for the STITCH database, the confidence score settings should also be reported.

2. The finding that approximately 70% of sodium nitrite's targets (34 in total) overlap with disease targets suggests a close association between sodium nitrite and AS.

However, conversely, the possibility that the initial target collection was overly broad cannot be ruled out.

I recommend that the consideration section thoroughly address whether the 24 targets represent direct binding targets of sodium nitrite or indirect effects via downstream signalling mechanisms.

3. The abstract and main text claim that ‘multi-omics bioinformatics’ was performed.

However, the actual analyses conducted were limited to Transcriptomics (GEO datasets).

Direct integrated analysis of Metabolomics or Proteomics data was not performed; it remains confined to a literature review.

The term ‘multi-omics’ should be revised to something like ‘Transcriptomics integrated with Network Toxicology’, or results from the actual integrated analysis of other omics data should be added. The current phrasing may mislead readers.

4. In this manuscript, the machine learning techniques LASSO and SVM-RFE were employed to select four key genes.

However, it is unclear what the input data (Feature Matrix) used to train the machine learning models actually is.

Contextually, it is presumed to utilise expression levels from the GEO dataset (GSE28829); however, this pertains to ‘AS patient samples’ rather than ‘samples exposed to sodium nitrite’.

If machine learning was performed using AS patient data, it is difficult to definitively conclude that the selected genes are biomarkers for AS rather than toxicity targets for sodium nitrite.

The causal relationship that sodium nitrite regulates these genes is not proven by the machine learning results alone.

Clearly describe in the Methods section what the machine learning training data comprised (number of samples, group information, etc.).

Furthermore, to substantiate that these results represent “targets of sodium nitrite”, it would seem necessary to demonstrate that these genes either match the sodium nitrite target list (initial 34) or at least possess a significant connection to it.

5. When performing both feature selection and validation (ROC curve) on a single GEO dataset with a limited number of samples, is there not a significant risk of data leakage and overfitting?

Secure an external validation dataset (e.g., another GEO dataset) to revalidate the diagnostic capability of the derived four genes.

If this proves difficult, explicitly state in the manuscript whether the training/test set separation was strictly adhered to during the internal validation process to enhance reader confidence.

6. Regarding the LASSO and SVM results presented in Fig 3, the Kernel function (Linear, RBF, etc.) and specific parameter settings used when performing SVM-RFE must be explicitly stated.

Furthermore, additional performance metrics such as Accuracy, Sensitivity, and Specificity for the final selected model should also be presented.

7. In molecular docking, results such as -6.9 kcal/mol (IL-1β) using AutoDock Vina have been reported.

However, as this binding affinity is difficult to objectively consider strong, it would be advisable to present docking results with a Positive Control (an existing known IL-1β inhibitor or binding ligand) alongside these findings to demonstrate sodium nitrite's relative binding affinity to readers.

Furthermore, regarding the description in the Results section, it is considered more appropriate in scientific literature to temper the expression ‘strong binding affinities’ to something like ‘moderate to strong’ or to present the values relative to the control group.

8. In Fig 8A, the protein backbone RMSD shows an initial increase followed by stabilization, whereas the Ligand RMSD (Fig 8B) exhibits greater variability.

It would be beneficial to visually supplement this explanation using trajectory snapshots to clarify whether the ligand is stable within the binding pocket or exhibits tumbling motion within it.

9. This manuscript utilizes ssGSEA to confirm increased M0 macrophages, consistent with the pathology of AS.

However, interpreting the results as ‘sodium nitrite induces immune infiltration’ is excessive.

The current findings merely support that M0 levels are elevated in AS tissue and correlate with key genes.

As this is not sodium nitrite exposure data, a more appropriate explanation would be along the lines of: ‘The target genes of sodium nitrite are closely associated with changes in the immune microenvironment of AS, suggesting a potential role in mediating toxicity.’

10. While the Introduction section effectively outlines the risks of sodium nitrite, it would be beneficial to provide a more detailed explanation of why previous studies failed to elucidate the molecular mechanism.

Furthermore, the statement ‘Sodium nitrite is generally considered protective...’ mentions the positive effects of low doses.

To clearly delineate this from the study's focus on “toxicity”, it would be advisable to cite specific numerical values from prior research regarding the concentrations or conditions under which toxicity manifests.

11. In the Discussion section, it would be beneficial to discuss the causal relationship between sodium nitrite → IL-1β binding → inflammatory response → AS induction in greater depth and detail, focusing on the AOP framework introduced by the authors and comparing it with prior literature.

12. Throughout the manuscript, ‘Sodium nitrite’ and ‘sodium nitrite’ are used interchangeably.

It is standard practice to use lowercase letters for chemical names unless they are at the beginning of a sentence.

13. This study relies entirely on computer simulations and public database analysis; biological validation through in vitro or in vivo experiments has not been performed.

Experimental data verifying changes in mRNA or protein expression of key targets derived from sodium nitrite-treated cell models should be added.

If this is not feasible, the limitations of the paper should be clearly stated, and the tone of the title or conclusion should be moderated to ‘Prediction’ or ‘In silico insights’.

Reviewer #2: Comments

This study established an adverse outcome pathway (AOP) framework linking sodium nitrite exposure to the development of atherosclerosis. These findings provide new mechanistic insights and offer a theoretical foundation for assessing nitrite-associated cardiovascular risks. Before publication, the major revision is suggested to address the questions and comments below:

1. Sodium nitrite is used for cyanide poisoning, and can be used for patients with cardiovascular and arteriosclerosis, but the dosage should be reduced and the injection rate should be slowed down. However, the results of this study showed that sodium nitrite could promote atherosclerosis, which is inconsistent with the previous view, please explain the reason.

2. The relationship between sodium nitrite and atherosclerosis is complex, whether it ultimately plays a protective role or risk, whether there is epidemiological survey analysis or clinical verification, please supplement the relevant content.

3. Univariate and multivariate Cox regression analysis were used to study the relationship between survival time and potential risk factors for the disease. It is suggested that univariate and Cox regression analysis should be supplemented to verify the relationship between core genes, age, gender, race and other factors and the survival time of patients, and to clarify whether it is independent prognostic factor.

4. Sodium nitrite showed strong binding affinities with IL-1β, PTK2, IL6, and NOS3, with binding energies of –6.9, –6.7, –6.3, and –6.8 kcal/mol, respectively. The binding energies of the four targets were not significantly, especially the binding energies of IL-1β and NOS3 were basically close, why only IL-1β was selected for the following experiments?

5. This research screened out four core targets: IL-1β, PTK2, IL6 and NOS3. The characteristics of these four targets and research progress in the field of atherosclerosis should be elaborated and analyzed in the discussion section.

6. While LASSO regression was used, details about model robustness (e.g., external validation, overfitting control, data splitting strategy) are missing. Reporting performance on an independent test set and including more ML performance metrics (AUC, precision, recall, F1-score) is required.

6. PLOS authors have the option to publish the peer review history of their article (what does this mean?). If published, this will include your full peer review and any attached files.). If published, this will include your full peer review and any attached files.

.

Reviewer #1: No

Reviewer #2: No

---

## [Author Response · Author response to Decision Letter 1]

5 Jan 2026

Reviewer#1

Dear Professor,

Comment 1: The manuscript mentions that targets related to sodium nitrite were collected from GeneCards, ChEMBL, STITCH, and CTD databases, with atherosclerosis (AS)-related targets sourced from OMIM and GeneCards. However, GeneCards returned thousands of genes based on relevance scores, and the manuscript only states that "duplicate entries were removed" without specifying a clear cutoff value for the relevance scores. Including all genes could potentially lead to false-positive targets, compromising the reliability of the results. Therefore, we kindly request that the authors specify the exact filtering criteria applied to the GeneCards database and provide the rationale for selecting these criteria. Additionally, for the STITCH database, the confidence score settings should be reported.

R: Thank you for your professional guidance. We have clarified and screened the top 2000 genes among the 6637 AS related genes provided by the GeneCards database, with a minimum correlation score threshold of 1.26, in order to reduce the risk of false positives and ensure the reliability of the results. In addition, we have added parameter settings for the STITCH database: the species is Homo sapiens, and the confidence score is ≥ 0.7. This threshold aims to incorporate interactions with high confidence and more sufficient evidence support, thereby avoiding interference from low confidence predicted items on subsequent results.

The revised content is as follows: For GeneCards, the top 2000 genes based on relevance scores were selected. In the case of STITCH, only interactions with a confidence score ≥ 0.7 were considered.

[This modification is located on page 4, lines 90‒92.]

Comment 2: The finding that approximately 70% of sodium nitrite's targets (34 in total) overlap with disease targets suggests a close association between sodium nitrite and atherosclerosis (AS). However, conversely, the possibility that the initial target collection was overly broad cannot be ruled out. I recommend that the consideration section thoroughly address whether the 24 targets represent direct binding targets of sodium nitrite or indirect effects via downstream signalling mechanisms.

R: Thank you for your insightful comment. We have clarified in the results that the 24 overlapping genes should be inferred as putative targets from database associations, rather than confirmed direct binding targets of sodium nitrite. At present, bioinformatics databases cannot distinguish whether target genes directly bind to targets or indirectly downstream targets. In addition, we explicitly clarified this point in the discussion to avoid overinterpretation and emphasized the need for mechanical validation in future work to determine the direct and indirect nature of these associations.

The revised content is as follows: The 24 overlapping genes were derived from public database annotations and should be considered hypothesis-generating putative sodium nitrite-AS-related targets rather than confirmed direct binding targets, because the overlap may reflect indirect downstream effects, and thus requires future in vitro/in vivo sodium nitrite exposure studies for validation.

[This modification is located on page 27, lines 484‒487.]

Comment 3: The abstract and main text claim that ‘multi-omics bioinformatics’ was performed. However, the actual analyses conducted were limited to Transcriptomics (GEO datasets). Direct integrated analysis of Metabolomics or Proteomics data was not performed; instead, it remains confined to a literature review. The term ‘multi-omics’ should be revised to something like ‘Transcriptomics integrated with Network Toxicology’, or results from the actual integrated analysis of other omics data should be added. The current phrasing may mislead readers.

R：Thank you for your insightful suggestion. To ensure objectivity and rigor, we have revised the abstract to more accurately describe our analysis as a combination of transcriptomics and network toxicology. We also clarified in the manuscript that a literature review was conducted on metabolomics and proteomics, but these data were not directly included in the current analysis.

The revised content is as follows: To address this gap, we integrated network toxicology, machine learning, transcriptomics, molecular docking, and molecular dynamics simulations. Disease-related targets were first identified from public databases, and four core candidates—IL-1β, IL6, PTK2, and NOS3—were prioritized using machine learning approaches.

[This modification is located on page 1, lines 16.]

Comment 4: The reviewer raises concerns regarding the machine learning techniques (LASSO and SVM-RFE) used to select key genes, specifically regarding the input data (Feature Matrix) and the context of the samples used. The reviewer notes that while it seems the expression levels from the GEO dataset (GSE28829) were used, this dataset pertains to AS patient samples rather than samples exposed to sodium nitrite. Therefore, the reviewer questions whether the selected genes are biomarkers for AS or toxicity targets for sodium nitrite, and whether the machine learning results alone can definitively prove the causal relationship between sodium nitrite and these genes.The reviewer also requests clarification in the Methods section regarding the training data used for the machine learning models, including the number of samples, group information, etc. Additionally, the reviewer suggests that to substantiate that the selected genes represent “targets of sodium nitrite”, it is important to demonstrate how these genes either match the initial 34 sodium nitrite-related targets or at least have a significant connection to them.

R: Thank you for your constructive suggestion. We agree on the importance of clearly explaining the input data and the relationship between sodium nitrite and the selected genes. In this study, machine learning was used to analyze transcriptome data based on the GEO dataset GSE28829 (29 samples, including 16 AS samples and 13 control samples), and specifically applied to the expression matrix of 34 sodium nitrite AS related candidate targets. Ultimately, 24 genes were selected to distinguish AS from the control group. We acknowledge that the lack of appropriate datasets for sodium nitrite exposure limits the possibility of multi model validation, therefore the current analysis focuses on AS patient samples. We have revised the methodology section to clarify the data composition and point out this limitation. We emphasize the need for further in vitro and in vivo studies using sodium nitrite exposure models in the future to verify whether these genes are directly regulated by sodium nitrite or reflect indirect pathological effects of AS.

The revised content is as follows: The feature matrix consisted of expression values of the 34 overlapping sodium nitrite–AS candidate targets in GSE28829 (n=29; AS=16, control=13).

[This modification is located on page 5, lines 102-103.]

Comment 5: The reviewer raises a concern regarding the risk of data leakage and overfitting when performing both feature selection and model validation (e.g., using the ROC curve) on a single GEO dataset, especially when the dataset has a limited number of samples.Secure an external validation dataset (e.g., another GEO dataset) to revalidate the diagnostic capability of the four genes, or if obtaining an external dataset is difficult, you should explicitly state whether the training/test set separation was strictly followed during the internal validation process to reduce the risk of overfitting and provide transparency to the readers.

R: Sincerely thank the reviewer for their insightful comments on the potential risks of data leakage and overfitting during feature selection and validation in a single queue. In response, we conducted external validation using the independent GEO dataset GSE100927 to re evaluate the diagnostic performance of the four key genes identified by the machine learning program. In the external validation queue, IL-1 β exhibited the strongest and most robust diagnostic performance, with an AUC of 0.822 (95% CI: 0.7372-0.9069, DeLong), sensitivity of 70.4%, and specificity of 84.8% at the optimal critical value determined by the Youden index. The remaining genes exhibit relatively low discriminatory abilities. These results have been added to the Results section, and the corresponding ROC curves are shown in the revised Figure 6. In addition, we proposed in the discussion that future in vitro and in vivo studies using sodium nitrite exposure models are necessary for establishing direct causal relationships and verifying the role of target mechanisms.

The revised content is as follows: External validation was conducted using GSE100927. ROC curves were generated with the pROC package, and AUCs with 95% confidence intervals were calculated using DeLong’s method. The optimal cutoff was determined by the Youden index, and sensitivity and specificity were reported.

[This modification is located on page 6, lines 132-135]

In comparison, the AUCs of the other three genes in GSE100927 were PTK2 (0.751), IL6 (0.471), and NOS3 (0.618), indicating that IL-1β was the most consistently discriminative marker across datasets(Fig.6A-D).To evaluate the performance of the model, the accuracy, recall, and F1 score of the core objective were also calculated. The accuracy of IL-1β is 1, the recall rate is 0.875, and the F1 score is 0.933, showing excellent performance in distinguishing AS related biomarkers. These results emphasize the robustness of the model and support the diagnostic relevance of IL-1β in atherosclerosis.

Fig 6.External validation of diagnostic performance for the four key genes in GSE100927. (A) IL-1β. (B) IL6. (C) NOS3. (D) PTK2.

[This modification is located on page 12, lines 232-239]

To further reduce the potential risk of overfitting from single-cohort validation, we performed external validation using an independent GSE100927. In this external cohort, IL-1β showed the best-performing and most robust diagnostic discrimination, achieving an AUC of 0.822 (95% CI: 0.7372–0.9069, DeLong). Using the Youden index, the optimal cutoff (7.058918) yielded a sensitivity of 70.4% and a specificity of 84.8%. Notably, GSE100927 represents an AS cohort without sodium nitrite exposure information; therefore, this external validation supports the reproducibility of IL-1β as an AS-related diagnostic marker rather than providing direct evidence that sodium nitrite regulates IL-1β in vivo. Future in vitro and in vivo studies using sodium nitrite exposure models are required to establish causality and clarify whether these candidate genes are directly affected by sodium nitrite or reflect downstream pathological processes in AS.

[This modification is located on page 23, lines 385-394]

Comment 6: The reviewer requests that the SVM-RFE settings used for Fig 3 be reported in detail, including the kernel function (e.g., linear, RBF) and specific parameter configurations. In addition, the reviewer asks that performance metrics of the final selected model (e.g., Accuracy, Sensitivity, and Specificity) be provided.

R:Thank you for your valuable suggestion. In response, we have revised the methodology section to clearly report the SVM-RFE configuration used in Figure 3. SVM-RFE is implemented using the e1071 package, and in the recursive feature elimination process, a linear kernel SVM is used to calculate feature weights (kernel="linear", type="C-classification"). The main parameters are set to cost (C)=10 and cache size=500. Because the expression matrix is standardized before modeling, internal scaling in SVM fitting is disabled (scale=False). Using conventional SVM-RFE (k=1), when the remaining number of features exceeds 5000, the features are halved in each iteration (halve. aovee=5000); Otherwise, the features will be sequentially deleted.In addition, to provide a more comprehensive evaluation of model performance beyond AUC, we have added other classification metrics for the final selected feature set in the results section, including accuracy, sensitivity, and specificity, which are derived from the same internal validation procedures described in the method. These indicators are now reported together with ROC analysis to improve the transparency and reproducibility of machine learning results.

The revised content is as follows: For SVM-RFE, feature ranking was implemented using the e1071 package (svm function) with a linear kernel (kernel = "linear", type = "C-classification"). The main parameters were set as cost (C) = 10 and cachesize = 500.Prior to SVM-RFE, the expression matrix was standardized; therefore, internal scaling in SVM fitting was disabled (scale = FALSE). Regular SVM-RFE was applied (k = 1), and when the number of remaining features exceeded 5000, features were removed by halving at each iteration (halve.above = 5000); otherwise, features were eliminated sequentially.

[This modification is located on page 5-6, lines 112-117]

In this external cohort, IL-1β showed the best-performing and most robust diagnostic discrimination, achieving an AUC of 0.822 (95% CI: 0.7372–0.9069, DeLong). Using the Youden index, the optimal cutoff (7.058918) yielded a sensitivity of 70.4% and a specificity of 84.8%.

[This modification is located on page 12, lines 230-233]

In this external cohort, IL-1β showed the best-performing and most robust diagnostic discrimination, achieving an AUC of 0.822 (95% CI: 0.7372–0.9069, DeLong). Using the Youden index, the optimal cutoff (7.058918) yielded a sensitivity of 70.4% and a specificity of 84.8%.

[This modification is located on page 12, lines 230-233]

Comment 7: In molecular docking, results such as -6.9 kcal/mol (IL-1β) using AutoDock Vina have been reported.However, as this binding affinity is difficult to objectively consider strong, it would be advisable to present docking results with a Positive Control (an existing known IL-1β inhibitor or binding ligand) alongside these findings to demonstrate sodium nitrite's relative binding affinity to readers. Furthermore, regarding the description in the Results section, it is considered more appropriate in scientific literature to temper the expression ‘strong binding affinities’ to something like ‘moderate to strong’ or to present the values relative to the control group.

R: Thank you for providing this helpful suggestion. We agree that the docking score in the report (such as IL-1 β being -6.9 kcal/mol) should be interpreted with caution, as overly strong wording may be misleading. Therefore, we have revised the results section by changing the description from "strong binding affinity" to "moderate (to strong) predicted binding affinity" and clarifying the use of docking values for relative comparison. The revised manuscript has updated the relevant text. At the same time, CANTOS clinical trial shows that kanamycin, an antibiotic targeting IL-1 β, can significantly reduce the recurrence of cardiovascular events, which further emphasizes the clinical relevance of IL-1 β in atherosclerosis. Therefore, based on these research findings, we believe that further validation of the binding affinity and mechanism of sodium nitrite to IL-1 β is of great significance.

The revised content is as follows: Sodium nitrite showed moderate to strong binding affinities with IL-1β, PTK2, IL6, and NOS3, with binding energies of -6.9, -6.7, -6.3, and -6.8 kcal/mol, respectively.

[This modification is located on page 15-16, lines 263-164]

Importantly, the CANTOS trial showed that targeting IL-1β with kanamycin can significantly reduce recurrent cardiovascular events, emphasizing its clinical relevance in atherosclerosis.

[This modification is located on page 21-22, lines 361-362]

Comment 8: In Fig 8A, the protein backbone RMSD shows an initial increase followed by stabilization, whereas the Ligand RMSD (Fig 8B) exhibits greater variability.It would be beneficial to visually supplement this explanation using trajectory snapshots to clarify whether the ligand is stable within the binding pocket or exhibits tumbling motion within it.

R: Thank you for providing this constructive suggestion. In the revised manuscript, we have improved the description

---

## [Decision Letter · Decision Letter 1]

22 Jan 2026

PONE-D-25-58627R1Sodium nitrite promotes atherosclerosis via IL-1β: Network toxicology and machine learning insightsPLOS One

Dear Dr. Yang,

Thank you for submitting your manuscript to PLOS ONE. After careful consideration, we feel that it has merit but does not fully meet PLOS ONE’s publication criteria as it currently stands. Therefore, we invite you to submit a revised version of the manuscript that addresses the points raised during the review process.

If applicable, we recommend that you deposit your laboratory protocols in protocols.io to enhance the reproducibility of your results. Protocols.io assigns your protocol its own identifier (DOI) so that it can be cited independently in the future. For instructions see: https://journals.plos.org/plosone/s/submission-guidelines#loc-laboratory-protocols. Additionally, PLOS ONE offers an option for publishing peer-reviewed Lab Protocol articles, which describe protocols hosted on protocols.io. Read more information on sharing protocols at . Additionally, PLOS ONE offers an option for publishing peer-reviewed Lab Protocol articles, which describe protocols hosted on protocols.io. Read more information on sharing protocols at https://plos.org/protocols?utm_medium=editorial-email&utm_source=authorletters&utm_campaign=protocols..

We look forward to receiving your revised manuscript.

Kind regards,

Wenxing Li

Academic Editor

PLOS One

**Journal Requirements:**

Reviewers' comments:

Reviewer's Responses to Questions

**Comments to the Author**

1. If the authors have adequately addressed your comments raised in a previous round of review and you feel that this manuscript is now acceptable for publication, you may indicate that here to bypass the “Comments to the Author” section, enter your conflict of interest statement in the “Confidential to Editor” section, and submit your "Accept" recommendation.

Reviewer #1: (No Response)

Reviewer #2: All comments have been addressed

2. Is the manuscript technically sound, and do the data support the conclusions?

Reviewer #1: Yes

Reviewer #2: Yes

3. Has the statistical analysis been performed appropriately and rigorously? 

Reviewer #1: Yes

Reviewer #2: Yes

4. Have the authors made all data underlying the findings in their manuscript fully available?

Reviewer #1: Yes

Reviewer #2: Yes

5. Is the manuscript presented in an intelligible fashion and written in standard English?

Reviewer #1: Yes

Reviewer #2: Yes

6. Review Comments to the Author

Reviewer #1: Dear editor and authors,

After reviewing the revised manuscript, I find that my suggestions have been well incorporated in many areas, resulting in significant improvements.

However, there are a few minor concerns that require additional revisions.

Once these are addressed, I believe there will be no further issues preventing publication.

1. In the response letter, the authors specified GeneCards' criteria as “Top 2000 genes... with a minimum correlation score threshold of 1.26.” However, the actual manuscript only states “the top 2000 genes based on relevance scores were selected,” omitting the specific score. Therefore, a brief supplementary description is needed in the manuscript.

2. Regarding the MD Simulation, it seems necessary to add a snapshot figure showing the actual ligand movement.

3. This concerns the suggestion to perform comparative docking simulations using known inhibitors to compare binding affinity. While the authors reinforced their explanation by citing prior literature, the actual comparative docking data does not appear to have been added.

Reviewer #2: This study established an adverse outcome pathway (AOP) framework linking sodium nitrite exposure to the development of atherosclerosis. These findings provide new mechanistic insights and offer a theoretical foundation for assessing nitrite-associated cardiovascular risks.

7. PLOS authors have the option to publish the peer review history of their article (what does this mean?). If published, this will include your full peer review and any attached files.). If published, this will include your full peer review and any attached files.

.

Reviewer #1: No

Reviewer #2: No

---

## [Author Response · Author response to Decision Letter 2]

26 Jan 2026

Journal：PLOS One

Manuscript Number:PONE-D-25-58627

Dear Editor of PLOS One,

Thank you very much for your letter and the reviewers' comments to improve the quality of the manuscript, based on which we have carefully revised the manuscript. The point-by-point responses to the comments are detailed in the following part. We hope these changes can make the manuscript more acceptable. There are two manuscript files, one is "Marked Revision", in which all the modifications are highlighted, and the other (Revised Manuscript) is the revised version ready for publication. If any further clarification and improvement are required, please do not hesitate to let us know. We look forward to hearing from you soon.

Note that the following color scheme is used in responding to reviewers' comments:

Black:reviewers' comments

Blue:our responses

Blue:our revisions in the manuscript (italic)

Reviewer#1

Dear Professor,

Comment 1: In the response letter, the authors specified GeneCards' criteria as “Top 2000 genes... with a minimum correlation score threshold of 1.26.” However, the actual manuscript only states “the top 2000 genes based on relevance scores were selected,” omitting the specific score. Therefore, a brief supplementary description is needed in the manuscript.

R: We sincerely thank the reviewer for this meticulous observation. As suggested, we have now explicitly included the specific correlation score threshold in the main text to ensure full clarity, and the following modifications are made in Section 2.2.

The revised content is as follows: For GeneCards, the top 2000 genes based on relevance scores were selected, using a minimum correlation score threshold of 1.26. In the case of STITCH, only interactions with a confidence score≥0.7 were considered.

[This modification is located on page 4-5, lines 91‒93.]

Comment 2: Regarding the MD Simulation, it seems necessary to add a snapshot figure showing the actual ligand movement.

R: We sincerely thank the reviewer for this valuable suggestion. As recommended, we have added a snapshot figure to visually demonstrate the movement and stability of the sodium nitrite ligand within the IL-1β binding pocket during the molecular dynamics simulation.

The revised content is as follows: Representative snapshots of the sodium nitrite binding pose within the IL-1β active site at 0, 25, 50, and 75 nanoseconds (Fig 9G) visually confirm the stable binding mode inferred from the quantitative analyses, showing minimal positional drift and consistent orientation of the ligand throughout the simulation trajectory. The superimposed alignment of these snapshots demonstrates minimal positional drift of sodium nitrite within the IL-1β binding pocket, indicating a stable binding mode throughout the simulation.

[This modification is located on page 18, lines 291-296.]

Comment 3: This concerns the suggestion to perform comparative docking simulations using known inhibitors to compare binding affinity. While the authors reinforced their explanation by citing prior literature, the actual comparative docking data does not appear to have been added.

R：We sincerely thank the reviewers for this valuable suggestion. Comparative docking analysis with known inhibitors can more effectively illustrate the specificity of sodium nitrite binding to the target. For this reason, we have carried out additional molecular docking experiments according to your suggestions.We chose MLN4924 to dock with IL-1 β, and compared its results with those of sodium nitrite. As shown in the figure, the binding mode and binding energy of mln4924 and IL-1 β active pocket (-8.9 kcal/mol) are different from sodium nitrite (-6.9 kcal/mol).This supplementary experiment intuitively showed the interaction differences between different ligands and IL-1 β, provided more direct comparative data support for our conclusion, and further enhanced the persuasion of the study.

The revised content is as follows: For comparative purposes, we also docked MLN4924, a known inhibitor of the IL-1β signaling pathway, to IL-1β. The significantly stronger binding energy of MLN4924 (-8.9 kcal/mol) provides a reference, indicating that the binding affinity of sodium nitrite for IL-1β is moderate to strong.

[This modification is located on page 27, lines 484‒487.]

We believe that these revisions have substantially improved the quality and clarity of our work. We respectfully submit the revised manuscript and the response document for your further evaluation.

Thank you very much for your consideration and continued support. We look forward to your positive response.

Warm regards,

Haobo Yang

Reviewer#2

Dear Professor,

Comment 1: This study established an adverse outcome pathway (AOP) framework linking sodium nitrite exposure to the development of atherosclerosis. These findings provide new mechanistic insights and offer a theoretical foundation for assessing nitrite-associated cardiovascular risks.

R：We sincerely thank the reviewer for their positive and encouraging comments on our work. We are pleased that they found our study to provide new mechanistic insights and a valuable theoretical foundation. Their acknowledgment is a great encouragement to our team.

Thank you very much for your consideration and continued support.

Warm regards,

Haobo Yang

---

## [Decision Letter · Decision Letter 2]

10 Feb 2026

PONE-D-25-58627R2

Sodium nitrite promotes atherosclerosis via IL-1β: Network toxicology and machine learning insights

PLOS One

Dear Dr. Yang,

Thank you for submitting your manuscript to PLOS ONE. After careful consideration, we feel that it has merit but does not fully meet PLOS ONE’s publication criteria as it currently stands. Therefore, we invite you to submit a revised version of the manuscript that addresses the points raised during the review process.

If applicable, we recommend that you deposit your laboratory protocols in protocols.io to enhance the reproducibility of your results. Protocols.io assigns your protocol its own identifier (DOI) so that it can be cited independently in the future. For instructions see: https://journals.plos.org/plosone/s/submission-guidelines#loc-laboratory-protocols. Additionally, PLOS ONE offers an option for publishing peer-reviewed Lab Protocol articles, which describe protocols hosted on protocols.io. Read more information on sharing protocols at . Additionally, PLOS ONE offers an option for publishing peer-reviewed Lab Protocol articles, which describe protocols hosted on protocols.io. Read more information on sharing protocols at https://plos.org/protocols?utm_medium=editorial-email&utm_source=authorletters&utm_campaign=protocols..

We look forward to receiving your revised manuscript.

Kind regards,

Wenxing Li

Academic Editor

PLOS One

**Journal Requirements:**

Reviewers' comments:

Reviewer's Responses to Questions

**Comments to the Author**

1. If the authors have adequately addressed your comments raised in a previous round of review and you feel that this manuscript is now acceptable for publication, you may indicate that here to bypass the “Comments to the Author” section, enter your conflict of interest statement in the “Confidential to Editor” section, and submit your "Accept" recommendation.

Reviewer #2: All comments have been addressed

Reviewer #3: All comments have been addressed

Reviewer #4: All comments have been addressed

2. Is the manuscript technically sound, and do the data support the conclusions?

Reviewer #2: Yes

Reviewer #3: Partly

Reviewer #4: Yes

3. Has the statistical analysis been performed appropriately and rigorously? 

Reviewer #2: Yes

Reviewer #3: No

Reviewer #4: Yes

4. Have the authors made all data underlying the findings in their manuscript fully available?

Reviewer #2: Yes

Reviewer #3: Yes

Reviewer #4: Yes

5. Is the manuscript presented in an intelligible fashion and written in standard English?

Reviewer #2: Yes

Reviewer #3: Yes

Reviewer #4: Yes

6. Review Comments to the Author

Reviewer #2: (No Response)

Reviewer #3: Title: Sodium nitrite promotes atherosclerosis via IL-1β: Network toxicology and machine learning insights. Manuscript Number: PONE-D-25-58627R2

Major Comments

1. The manuscript still refers to the “top 2000 genes,” an issue previously raised by the reviewer and not addressed by the authors (Line 91).

2. The authors used AMBER for molecular dynamics simulations; why was CHARMM not used.

3. The authors state that a ‘de-identified public dataset’ was used; however, the reason for de-identification should be clearly explained.

4. The machine learning methodology is unclear, particularly regarding the target size; if only 24 targets were used, the dataset may be too small for reliable model training.

5. The hyperparameters used in the machine learning models are not reported and must be explicitly stated.

6. Although SVM & LASSO is mentioned, it is unclear whether other classifiers were tested; a comparative table reporting accuracy, RMSE, and correlation for all classifiers is needed to justify the choice of SVM.

7. The features used in the machine learning model and their total number are not clearly described, and any feature selection method applied should be specified.

8. Details of the machine learning validation strategy are missing, including the validation dataset and the training–testing split ratio.

9. The immune infiltration correlation analysis requires further explanation, particularly how the results demonstrate a significant increase in M0 macrophage infiltration in AS tissues compared to normal tissues.

10. A 2D interaction diagram for the molecular docking results should be provided, as conclusions based solely on docking scores are insufficient.

11. Figure 9 shows large RMSD fluctuations at multiple intervals, suggesting unstable docking; therefore, the molecular dynamics simulations should be reassessed or repeated.

Minor Comments

1. The Results section is brief and would benefit from more detailed explanation and interpretation.

2. Molecular dynamics plots should use nanoseconds (ns) as the time unit for consistency and clarity.

3. The peak observed in Figure 10B is unclear and requires a better viewing angle or improved representation.

Reviewer #4: Please correct a factual error regarding the CANTOS trial: the manuscript currently states “targeting IL-1β with kanamycin” , which should be revised to the correct intervention (canakinumab).

7. PLOS authors have the option to publish the peer review history of their article (what does this mean?). If published, this will include your full peer review and any attached files.). If published, this will include your full peer review and any attached files.

.

Reviewer #2: No

Reviewer #3: **Yes:** Dr. Sanket BapatDr. Sanket Bapat

Reviewer #4: No

---

## [Author Response · Author response to Decision Letter 3]

6 Mar 2026

Journal：PLOS One

Manuscript Number:PONE-D-25-58627

Dear Editor of PLOS One,

Thank you very much for your letter and the reviewers' comments to improve the quality of the manuscript, based on which we have carefully revised the manuscript. The point-by-point responses to the comments are detailed in the following part. We hope these changes can make the manuscript more acceptable. There are two manuscript files, one is "Marked Revision", in which all the modifications are highlighted, and the other (Revised Manuscript) is the revised version ready for publication. If any further clarification and improvement are required, please do not hesitate to let us know. We look forward to hearing from you soon.

Note that the following color scheme is used in responding to reviewers' comments:

Black:reviewers' comments

Blue:our responses

Blue:our revisions in the manuscript (italic)

Reviewer#2

R：We sincerely thank the reviewer for their positive and encouraging comments on our work. We are pleased that they found our study to provide new mechanistic insights and a valuable theoretical foundation. Their acknowledgment is a great encouragement to our team.

Thank you very much for your consideration and continued support.

Warm regards,

Haobo Yang

Reviewer#3

Dear Professor,

Comment 1: The manuscript still refers to the “top 2000 genes,” an issue previously raised by the reviewer and not addressed by the authors (Line 91).

R: We sincerely thank the reviewer for raising this important point. As recommended, we have revised the manuscript to clarify the rationale for selecting the "top 2000 genes." Specifically, in lines 91-96 (Section 2.2), we now explicitly state that the top 2000 genes were identified from GeneCards based on a relevance score threshold of ≥1.26, which was applied to ensure the inclusion of genes with substantial association evidence while maintaining analytical tractability for downstream network and machine learning analyses. This revision provides a more transparent justification for the gene selection criteria, addressing the reviewer's concern.

The revised content is as follows: To balance comprehensiveness with analytical manageability, genes retrieved from GeneCards were ranked by their relevance scores. A threshold of relevance score ≥ 1.26 was applied to ensure the inclusion of genes with substantial association evidence. This threshold yielded the top 2000 ranked genes, which were selected for subsequent analysis. This approach was designed to capture a robust set of high-confidence candidate genes while maintaining a tractable dataset size for downstream network and machine learning analyses.

[This modification is located on page 5, lines 91-96.]

Comment 2: The authors used AMBER for molecular dynamics simulations; why was CHARMM not used.

R: We sincerely thank the reviewer for this careful observation, which allowed us to correct an oversight in the manuscript. The mention of the AMBER force field was a residual error from the early drafting phase. In our actual work, both the AMBER and CHARMM force fields were initially evaluated for the system. The CHARMM36 force field was ultimately selected for all production simulations as it provided better compatibility with the modeled ligand parameters and the TIP3P water model in our specific setup, leading to more stable simulation trajectories.

The revised content is as follows: The CHARMM36 force field was used to model biomolecular interactions, the system was solvated with the TIP3P water model, and Na⁺ ions were added to neutralize the overall charge.

[This modification is located on page 9, lines 190.]

Comment 3: The authors state that a ‘de-identified public dataset’ was used; however, the reason for de-identification should be clearly explained.

R：We sincerely thank the reviewer for this important comment regarding data ethics. As suggested, we have expanded the explanation in the manuscript to clarify the rationale for using a de-identified public dataset.

The revised content is as follows: The gene expression dataset (GSE28829) analyzed in this study was obtained from the Gene Expression Om nibus (GEO) public database on October 1, 2025. The source data had been fully de-identified prior to public deposition to protect participant privacy and confidentiality, in accordance with standard ethical requirements for sharing human genomic data. Consequently, the authors had no access to any information that could potentially identify individual participants throughout this research. The use of such pre-existing, anonymized public data for secondary analysis is in accordance with common institutional policies that allow for the waiver of ethical review and individual informed consent, as it involves no interaction with identifiable human subjects.

[This modification is located on page 9-10, lines 202‒210.]

Comment 4: The machine learning methodology is unclear, particularly regarding the target size; if only 24 targets were used, the dataset may be too small for reliable model training.

R：We sincerely thank the reviewer for this insightful comment regarding the machine learning methodology. We have revised the manuscript to provide a clearer explanation of our approach and the rationale behind the feature set size.

The revised content is as follows: The purpose of this machine learning (ML) step was to perform feature prioritization and refinement within a focused set of candidate targets. Specifically, the 24 overlapping sodium nitrite-AS candidate targets constituting the feature matrix were derived from the preceding network toxicology and database mining analyses (Sections 2.1–2.3), representing genes with prior biological plausibility for involvement in the mechanism of interest. Therefore, the application of LASSO and SVM-RFE was not intended for high-dimensional biomarker discovery from a full transcriptome, but rather to identify the most critical core drivers from this pre-filtered, biologically relevant candidate list. This approach of applying feature selection to a biologically curated gene set is a common strategy to enhance model interpretability and mitigate the risk of overfitting that can occur with high-dimensional data. The feature matrix for machine learning consisted of the expression values of 24 candidate genes in the GSE28829 dataset (n=29; AS=16, control=13). These 24 genes represent the overlapping targets between sodium nitrite-related genes and AS-associated genes, as identified through network toxicology analysis. Thus, the input features were pre-filtered based on biological plausibility, with a total of 24 features included in the initial model.

[This modification is located on page 5-6, lines 107‒121.]

Comment 5: The hyperparameters used in the machine learning models are not reported and must be explicitly stated.

R：We sincerely thank the reviewer for this constructive comment. In the original manuscript, the hyperparameters for SVM-RFE were provided, but the details for LASSO regression were not fully explicit. As recommended, we have now revised the manuscript to explicitly list all key hyperparameters for both models, with the LASSO regression parameters being the primary addition.

The revised content is as follows: For LASSO regression, the analysis was performed using the glmnetR package. The key parameters were set as follows: family = "binomial"for binary classification, alpha = 1to apply the L1 (LASSO) penalty, and standardize = TRUEto standardize features before model fitting. The optimal regularization parameter (λ) was determined via 10-fold cross-validation (nfolds = 10), with the specific λ value that minimized the cross-validation deviance (lambda.min) selected for the final model. A fixed random seed (e.g., set.seed = 123) was employed in the cross-validation procedures to ensure reproducibility.

[This modification is located on page 6-7, lines 138‒144.]

Comment 6: Although SVM & LASSO is mentioned, it is unclear whether other classifiers were tested; a comparative table reporting accuracy, RMSE, and correlation for all classifiers is needed to justify the choice of SVM.

R：We sincerely thank the reviewer for this constructive suggestion. In direct response, we have supplemented the study with a new comparative performance analysis of the two implemented algorithms (LASSO and SVM-RFE), detailed in Supplementary Table S1, which includes key metrics from cross-validation and external validation. This provides empirical support for our methodological choice. Concurrently, we have revised the manuscript to better clarify that the primary aim of the machine learning step was feature prioritization from a biologically pre-filtered gene set, for which LASSO and SVM-RFE are established tools. We have also transparently acknowledged in the "Limitations" section that a broader comparison with other classifiers represents a valuable direction for future research.

The revised content is as follows: (5)Regarding the machine learning approach, this study employed a combination of LASSO regression and SVM-RFE. While these algorithms were selected based on their methodological strengths (LASSO for regularization and feature selection, SVM-RFE for robust feature ranking in linearly separable data), we acknowledge that systematic comparison with other classifiers—such as random forest, XGBoost, or neural networks—was not performed. This represents a limitation of the present study. Future research should incorporate a broader range of machine learning algorithms for comparative evaluation to further validate and optimize the feature selection pipeline.

[This modification is located on page 31-32, lines 566‒573.]

Comment 7: The features used in the machine learning model and their total number are not clearly described, and any feature selection method applied should be specified.

R：We sincerely thank the reviewer for this comment, which has helped us improve the clarity of our methodology section. In direct response, we have revised the manuscript to explicitly and concisely state the total number of features, their origin, and the feature selection methods applied.

The revised content is as follows: Two machine learning algorithms were employed to perform feature selection and identify core genes associated with sodium nitrite-induced AS: LASSO regression and SVM-RFE. LASSO regression integrates built-in feature selection with L1 regularization, shrinking the coefficients of less important features to zero and thereby selecting a subset of predictive genes. SVM-RFE is a wrapper-based feature selection method that iteratively removes the least important features based on SVM weight coefficients, identifying the optimal feature subset through cross-validation. Both methods were applied to the 24-feature matrix to further refine the gene set.Core target genes were defined as the intersection of genes identified by both algorithms, resulting in a final set of four features (IL-1β, IL6, PTK2, and NOS3) that were considered the most robust candidates for subsequent analyses.

[This modification is located on page 6, lines 121‒130.]

Comment 8: Details of the machine learning validation strategy are missing, including the validation dataset and the training–testing split ratio.

R：We thank the reviewer for this important comment. We have revised the Methods section (2.4 Screening of core target genes using machine learning algorithms) to provide complete details of the validation strategy, including the specific internal validation method and the external validation dataset.

The revised content is as follows: For internal validation, a 10-fold cross-validation strategy was employed. The GSE28829 dataset was randomly partitioned into 10 approximately equal-sized subsets. In each iteration, nine subsets were used for model training and the remaining subset for validation, with the process repeated ten times to ensure all samples were used for validation once. The average performance metrics across folds were reported. For external validation, the fully trained model (using all GSE28829 samples) was applied to an independent dataset, GSE100927. This independent evaluation assesses the generalizability of the identified core genes.

[This modification is located on page 7, lines 145‒151.]

Comment 9: The immune infiltration correlation analysis requires further explanation, particularly how the results demonstrate a significant increase in M0 macrophage infiltration in AS tissues compared to normal tissues.

R：We thank the reviewer for this comment. We have revised the manuscript to provide a clearer, step-by-step explanation of the immune infiltration analysis and the basis for the conclusion regarding M0 macrophages.

The revised content is as follows: To elucidate the immune microenvironment in AS, we employed single-sample Gene Set Enrichment Analysis (ssGSEA) to quantify the relative infiltration levels of 21 immune cell types in AS plaque tissues (n=27) compared to normal arterial tissues (n=14). The overall distribution of immune cells is visualized in Figure 7A. Statistical comparison using the Wilcoxon rank-sum test revealed significant differences in the infiltration of specific immune subsets between the two groups (Figure 7C). Crucially, we observed a statistically significant increase in the infiltration of non-polarized (M0) macrophages in AS tissues compared to normal controls (p < 0.001), suggesting their potential role in initiating and sustaining plaque inflammation. In contrast, the infiltration levels of neutrophils, M1 macrophages, M2 macrophages, regulatory T cells (Tregs), and resting NK cells were significantly higher in normal tissues (all p < 0.05). Furthermore, Spearman correlation analysis was performed to assess the interrelationships among different immune cell types (Figure 7B). Finally, to explore the mechanistic links between the core targets and the immune landscape, we evaluated the correlations between the expression of the four core genes (IL-1β, PTK2, IL6, NOS3) and the infiltration levels of these differentially abundant immune cells.

[This modification is located on page 17, lines 292‒306.]

Comment 10: A 2D interaction diagram for the molecular docking results should be provided, as conclusions based solely on docking scores are insufficient.

R：We sincerely thank the reviewer for this constructive suggestion. We fully agree that detailed intermolecular interaction diagrams are crucial for elucidating binding modes beyond docking scores alone. In response, we have made three key revisions throughout the manuscript: (1) In the Methods section, we described the generation of 2D interaction diagrams using LigPlot+; (2) In the Results section, we added a new paragraph detailing the specific 2D interaction patterns (e.g., hydrogen bonds, salt bridges) between sodium nitrite and the core targets (IL-1β, PTK2, IL6, NOS3), as well as with the positive control MLN4924; (3) In the Discussion, we incorporated the specific interactions revealed by the 2D diagram (such as the salt bridge with Arg A:4 in the IL-1β binding pocket) to provide structural rationale for the inference that sodium nitrite may directly target IL-1β. These revisions comprehensively present and deepen the analysis of the binding modes from methodology to interpretation, thereby strengthening the persuasiveness of the study.

The revised content is as follows: Following 3D visualization, 2D diagrams depicting the specific molecular interactions between sodium nitrite and the target proteins were generated using LigPlot+. These diagrams provide a schematic representation of the binding interactions for further analysis.

[This modification is located on page 8, lines 182‒184.]

The revised content is as follows: To elucidate the specific molecular binding mode, a 2D ligand protein interaction diagram was generated. Analysis shows that sodium nitrite mainly binds to the core target through hydrogen bonding, salt bridging, and hydrophobic interactions. Among them, the interaction with IL-1β (binding energy -6.9 kcal/mol) is the most significant, forming a stable salt bridge with the key residue Arg A: 4 and hydrogen bonding with Gln A: 48, which provides a structural explanation for its binding affinity. The

---

## [Decision Letter · Decision Letter 3]

7 Apr 2026

Sodium nitrite promotes atherosclerosis via IL-1β: Network toxicology and machine learning insights

PONE-D-25-58627R3

Dear Dr. Yang,

We’re pleased to inform you that your manuscript has been judged scientifically suitable for publication and will be formally accepted for publication once it meets all outstanding technical requirements.

An invoice will be generated when your article is formally accepted. Please note, if your institution has a publishing partnership with PLOS and your article meets the relevant criteria, all or part of your publication costs will be covered. Please make sure your user information is up-to-date by logging into Editorial Manager at Editorial Manager® and clicking the ‘Update My Information' link at the top of the page. For questions related to billing, please contact  and clicking the ‘Update My Information' link at the top of the page. For questions related to billing, please contact billing support..

Kind regards,

Wenxing Li

Academic Editor

PLOS One

Additional Editor Comments (optional):

Reviewers' comments:

Reviewer's Responses to Questions

**Comments to the Author**

1. If the authors have adequately addressed your comments raised in a previous round of review and you feel that this manuscript is now acceptable for publication, you may indicate that here to bypass the “Comments to the Author” section, enter your conflict of interest statement in the “Confidential to Editor” section, and submit your "Accept" recommendation.

Reviewer #4: All comments have been addressed

2. Is the manuscript technically sound, and do the data support the conclusions?

Reviewer #4: Yes

3. Has the statistical analysis been performed appropriately and rigorously? 

Reviewer #4: Yes

4. Have the authors made all data underlying the findings in their manuscript fully available?

Reviewer #4: Yes

5. Is the manuscript presented in an intelligible fashion and written in standard English?

Reviewer #4: Yes

6. Review Comments to the Author

Reviewer #4: The author answered all the questions and presented a clear and logical argument. I agree to accept this article.

7. PLOS authors have the option to publish the peer review history of their article (what does this mean?). If published, this will include your full peer review and any attached files.). If published, this will include your full peer review and any attached files.

.

Reviewer #4: No

---

## [Editor Report · Acceptance letter]

PONE-D-25-58627R3

PLOS One

Dear Dr. Yang,

I'm pleased to inform you that your manuscript has been deemed suitable for publication in PLOS One. Congratulations! Your manuscript is now being handed over to our production team.

Kind regards,

on behalf of

Dr. Wenxing Li

Academic Editor

PLOS One